# Sharper Analysis of Data Echoing and New Communication-Efficient Algorithm for Data Parallelism

## Abstract

Over the past decade, breakthroughs in both general-purpose and specialized hardware have propelled the success of large-scale machine learning. However, the advancements in general-purpose hardware are not keeping pace with those in specialized hardware. Consequently, operations conducted on the general-purpose hardware have become the primary performance bottleneck. Notably, data loading significantly lags behind the gradient computation during training. To address this issue, the technique of data echoing has been introduced in practice, whereby the current batch of samples is reused for gradient computation to minimize idle time while waiting for new data. However, this approach can lead to overfitting on the current batch, and it remains unclear whether convergence benefits from this practice. In this paper, we provide a sharper analysis on a stochastic formulation of data echoing and show that it obtains linear speedup proportional to the number of reuse times. Additionally, we investigate the impact of the communication bottleneck in data parallelism to data echoing, and propose a new communication-efficient data echoing algorithm via reducing the frequency of model averaging. We then show that it is possible to perform data echoing without additional communication cost with data parallelism. Finally, we perform empirical experiments to verify our analysis on the data echoing and the proposed efficient algorithm for data parallelism.

## 1 Introduction

From the introduction of AlexNet (Krizhevsky et al., 2009), which is often considered as a milestone of the modern deep learning era, to the recent surge in large foundation language models such as OpenAI's GPT-4 (OpenAI, 2024) and Google's Gemini (Gemini Team, 2024), the field of artificial intelligence has experienced rapid evolution through the training of increasingly large models. A key driving factor behind this is the improvement of the specialized hardware such as GPU (Nickolls et al., 2010) and TPU (Jouppi et al., 2017), which can perform highly parallelism matrix computations and enable fast gradient computation to the large-scale model. Meanwhile, CPU-bound operations have emerged as a major performance bottleneck. As datasets cannot fit entirely into the memory of GPUs/TPUs, data must be loaded dynamically during training. However, the speed of data loading cannot keep pace with the computation demands. As illustrated in Figure 1, reaching the maximum FLOPS rate requires a specific level of arithmetic intensity, defined as the ratio of total FLOPS to the data movement needed to support those operations. In modern GPU training, the arithmetic intensity tends to be low due to slow data transfers, which hinders the system from achieving its peak FLOPS rate. To alleviate this bottleneck, the *data echoing* technique was proposed by Choi et al. (2019), whereby the current batch of examples in memory are reused for gradient computation to minimize idle time while waiting for new data. More specifically, given the following optimization problem: $\min_{x \in \mathbb{R}^d} f(x) := \mathbb{E}_{\xi \sim \mathcal{D}}[f(x;\xi)]$, and a minibatch $\mathcal{B}_t \sim \mathcal{D}$ of samples are loaded to the GPU memory at training step $t$, then we perform $M$ steps of gradient descent for this batch of data:

$$x_t^{(m+1)} = x_t^{(m)} - \eta \nabla f(x_t^{(m)}; \mathcal{B}_t), m \in [M] \tag{1}$$

where, $x_t^{(0)} = x_t$ and $x_{t+1} = x_t^{(M)}$. In Choi et al. (2019), the authors empirically demonstrate that data echoing reduces the number of loading operations required to achieve a target accuracy. However, although with empirical success, the theoretical benefits of the algorithm are still under-explored.

Data echoing is a type of biased stochastic gradient descent method, as the sequence of states $x_t^{(m)}, m \in [M]$ for a given time step $t$ are updated based on correlated (same) examples. As a result, the convergence analysis of data echoing is very challenging. The pioneer work by Agarwal et al. (2020) studied the convergence property of data echoing with stability analysis (Hardt et al., 2015), and showed that for convex problems with bounded gradient, the data-echoed algorithms achieve linear speedup *w.r.t.* the curvature term. But the dominant statistical term does not benefit from echoing. In this work, we move one step further and show that for a type of stochastic data-echoed algorithm, the statistical term also benefits from echoing and obtains linear speedup. More specifically, we demonstrate the linear speedup with a type of shifted state analysis to the gradient estimation bias. Recall that in Agarwal et al. (2020), the bias of $\nabla f(x_t^{(m)}; \mathcal{B}_t)$ is bounded *w.r.t.* $f(x_t^{(m)})$. Since

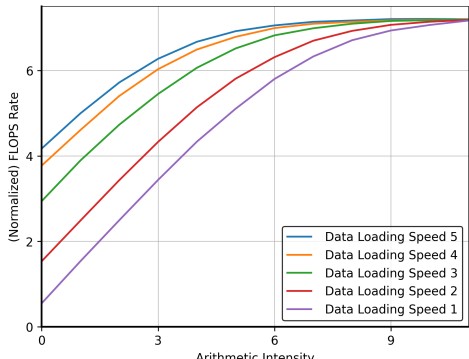

Figure 1: **(Normalized) FLOPS Rate *vs* Arithmetic Intensity under different Data Loading Speed levels**. For Data Loading Speed $i$, a higher value of $i$ indicates a faster loading speed. Refer to Section B.1 for more details on how this figure was generated.

$x_t^{(m)}$ and $\mathcal{B}_t$ are correlated, only the following bound can be got $\|\nabla f(x_t^{(m)}; \mathcal{B}_t) - \nabla f(x_t^{(m)})\| \le B$ given that the gradient of $f(x)$ is bounded by a constant $B$. In contrast, we perform a different analysis by bounding $\|\nabla f(x_{t-\tau}^{(m)}; \mathcal{B}_{t-\tau}) - \nabla f(x_t^{(m)})\|$. Given that $\mathcal{B}_{t-\tau}$ is independent from $x_t^{(m)}$ for sufficiently large $\tau$, $\mathbb{E}[\nabla f(x_t^{(m)}; \mathcal{B}_{t-\tau})] = \nabla f(x_t^{(m)})$. If $x_t^{(m)}$ is close to $x_{t-\tau}^{(m)}$, we can bound $\|\nabla f(x_{t-\tau}^{(m)}; \mathcal{B}_{t-\tau}) - \nabla f(x_t^{(m)})\|$ given the function $f$ is smooth.

Besides, we also investigate the data echoing algorithm in the data parallelism setting. To speed up the training of large-scale models, it is common to divide the examples across different nodes. In each training step, nodes first compute gradients in parallel and then average these gradients collectively. Note that the communication volume of the average step is at the order of $O(dK)$ ($d$ is the number of model parameters and $K$ is the number of nodes), which is comparable to that of the data loading. As a result, the benefit of doing data echo could disappear in the data parallelism setting, where the amount of communication among GPU/TPU nodes increase as more gradient steps are performed. To tackle this challenge, we propose a new communication-efficient data echoing algorithm in the data parallelism setting. More specifically, for each training step $t$, nodes only average gradients with a predefined probability $p^{(c)} > 0$. By carefully selecting the value of $p^{(c)}$, we show that it is possible to eliminate the communication overhead when we apply data echoing with data parallelism.

We summarize the **contributions** of our work as follows:

- We propose a stochastic formulation of data echoing and provide a sharper analysis which demonstrates that it achieves linear speedup with respect to the number of reuse steps after an initial burn-in period;

- We propose a new communication-efficient data echoing algorithm that enables data parallelism without introducing communication overhead;

- We propose a practical cosine diminishing schedule for data loading probability and validate its effectiveness through numerical experiments on a variety of benchmark datasets and deep learning models.

**Notations.** $\nabla f(x)$ denotes the first-order derivatives of the function $f(x)$ *w.r.t.* variable $x$. $\xi$ denotes a random sample and $\nabla f(x; \xi)$ is the stochastic estimate $\nabla f(x)$. $O(\cdot)$ is the big $O$ notation ($\tilde{O}(\cdot)$ omits the logarithmic terms, and hides the logarithmic terms. $\|\cdot\|$ denotes the $\ell_2$ norm for vectors and the spectral norm for matrices, respectively. $\langle \cdot, \cdot \rangle$ denotes the Euclidean inner product. [K] denotes the set of $\{1, 2, ..., K\}$. For a random variable $X$, $\mathbb{E}[X]$ denotes its expectation.

## 2 RELATED WORK

The term of data echoing is proposed in Choi et al. (2019), where authors consider the full training pipeline of a machine learning algorithm, including reading from the disk, shuffling the data, data augmentation, batching etc. Then in data echoing, each stage simply reuses the current available data instead of waiting for operations from the last stage to finish. In our work, we focus on the echoing of data batches similar to Agarwal et al. (2020). Agarwal et al. (2020) is a pioneered work in analyzing the convergence of data echoing. Its analysis consists of two parts: the first involves establishing bounds on the progress made with respect to an objective defined over a mini-batch, while the second part connects the decrease on a batch to the full dataset objective. The first part can be performed by standard regret analysis in optimization, while the second part utilizes the concept of uniform stability. Note that Agarwal et al. (2020) considers the set of convex optimization problems with bounded gradient. We study the general non-convex setting and do not require the bounded gradient assumption, instead we require a less restrictive bounded bias assumption (Even, 2023). Our analysis to data echoing is from the view of Markov chain gradient descent, while our communication-efficient data echoing algorithm is inspired by the Local Gradient Descent (*i.e.* FedAvg) (McMahan et al., 2017). We introduce some recent developments in these two directions below:

**Markov-Chain Gradient Descent.** Algorithms similar to stochastic gradient descent (Robbins & Monro, 1951) have achieved significant success in large-scale machine learning. Typically, these algorithms rely on access to *i.i.d.* examples to estimate the gradient. However, in some applications such as decentralized optimization (Johansson et al., 2007; 2010) and reinforcement learning (Sun et al., 2018), *i.i.d.* samples may not be readily available or may be costly to obtain. A well-studied alternative (Johansson et al., 2007; 2010; Sun et al., 2018; Doan et al., 2020; Even, 2023) involves sampling examples from a Markov chain. The example sequence in data echoing (Choi et al., 2019) can also be viewed as a Markov chain. The convergence of Markov chain gradient descent is analyzed under various assumptions to the properties of the involved chains. Under the ergodic assumption, Johansson et al. (2007) shows that $O(\epsilon^{-2})$ iterations are needed to achieve an $\epsilon$ error. Sun et al. (2018) shows for both reversible and non-reversible chains that $O(\epsilon^{-(1/(1-q))}), q \in (0.5, 1)$ iterations are needed for non-convex problems. Even (2023) improves over the analysis in Sun et al. (2018) by removing the bounded gradient assumption. Besides the convergence analysis, Wang et al. (2022) studied the generalization property of Markov chain gradient descent using the tool of algorithmic stability (Hardt et al., 2015); Adibi et al. (2024) studied the effect of delayed update to Markov chain gradient descent.

**Local Gradient Descent.** The idea of reducing communication cost by performing local gradient descent is used in Federated Learning (FL) (McMahan et al., 2017), which is a novel paradigm for performing machine learning tasks over distributed, privacy sensitive data. In FL, communication is the major performance bottleneck (Wen et al., 2017) and Local Gradient Descent is proposed to reduce the frequency of client-server communication. Our communication-efficient data echoing for data parallelism is inspired by the similar idea. Besides, various compression techniques (Wen et al., 2017; Lin et al., 2017; Stich et al., 2018; Karimireddy et al., 2019) can also be applied to reduce the communication cost and can be combined with data echoing (we leave this as a future work to explore).

## 3 STOCHASTIC DATA ECHOING AND A TIGHTER ANALYSIS

In this section, we introduce a new tighter convergence analysis to data echoing. More specifically, we consider a formulation of stochastic data echoing and view it as a type of Markov Chain gradient descent. We first introduce some preliminaries of Markov chain in the next subsection.

### 3.1 PRELIMINARIES OF MARKOV CHAIN

We consider the following finite-state time-homogeneous Markov Chain:

**Definition 3.1** (Adapted from Definition 1 in Agarwal et al. (2020)). *Let $P$ be an $N \times N$ real-valued matrix. A stochastic process $X = \{X_1, X_2, ...\}$ in a finite state space $[M]$ is a time-homogeneous Markov chain with transition matrix $P$ if for any $k \geq 0$, $i, j \in [M]$, $i_0, i_1, ..., i_{m-1} \in [M]$, the*

---

**Algorithm 1** Stochastic Data Echoing Algorithm

---

1: **Input**: $\{\eta_t\}$, learning rates; $\{p_t\}$, the probability of a example is replaced with a new one; $B$, the minibatch size; initialize the variable state as $x_1$.
2: **for** $t = 1$ to $T$ **do**
3:     Update the mini-batch $\mathcal{B}_t$ according to equation 4, *i.e.* replace each example with probability $p_t$
4:     Compute the stochastic gradient $d_t = \frac{1}{B} \sum_{z_t \in \mathcal{B}_t} \nabla f(x_t; z_t)$ with $|\mathcal{B}_t| = B$
5:     $x_{t+1} = x_t - \eta_t d_t$
6: **end for**
7: **Return**: $x_a$ chosen uniformly randomly from $\{x_t\}_{t=1}^T$

---

*following conditional independence property holds:*

$$\mathbb{P}(X_m = j | X_0 = i_0, ..., X_{m-1} = i) = \mathbb{P}(X_m = j | X_{m-1} = i) = P_{i.j}$$

*$P$ is the transition matrix of $X$ and the probability distribution $\pi$ is the stationary distribution of the stochastic process $X$ if $\pi P = \pi$.*

A Markov chain is *irreducible* if there exists some $m > 0$, s.t. $P_{i,j}^m > 0, i, j \in [N]$; a Markov chain is *aperiodic* if there exists $m_0 > 0$ such that for all $m > m_0$, $P_{i,j}^m > 0, i, j \in [N]$; a Markov chain is *reversible* if $\pi_i P_{i,j} = \pi_j P_{j,i}, i, j \in [N]$. Note that a irreducible, aperiodic Markov chain obtains a stationary distribution (Levin & Peres, 2017). We next define the *mixing time* of a Markov chain:

**Definition 3.2** (Adapted from Definition 2.2 in Even (2023)). *For $\epsilon > 0$, the mixing time $\tau_{mix}(\epsilon)$ is defined as follows: $\tau_{mix}(\epsilon) = \inf \{m \geq 1 | \forall \pi_0, d_{TV}(P^m \pi_0, \pi) \leq \epsilon\}$, where $d_{TV}$ is the total-variation distance.*

Note that $\tau_{mix}(\epsilon)$ has a logarithmic dependence over $\epsilon$, where we have $\tau_{mix}(\epsilon) \leq \tau_{mix} \log_2(\epsilon^{-1})$ (Even, 2023). Furthermore, if a Markov chain is reversible, its mixing time is closely related to the eigenvalue of the transition matrix. Denote the absolute spectral gap of $P$ as $\lambda_P = 1 - max_{\lambda \in Sp(P) \setminus \{1\}} |\lambda|$, where $Sp(P)$ is the spectrum of $P$. Then for reversible Markov chain, its mixing time satisfies $\tau_{mix}(\epsilon) \leq \lceil \lambda_P^{-1} \ln(\pi_{min}^{-1} \epsilon^{-1}) \rceil$ (Even, 2023), where $\pi_{min}$ denotes the minimum element of the stationary distribution $\pi$.

## 3.2 THE STOCHASTIC DATA ECHOING

We consider the following stochastic optimization objective in this work:

$$\min_{x \in \mathbb{R}^d} f(x) \coloneqq \mathbb{E}_{\xi \sim \mathcal{D}}[f(x; \xi)] = \frac{1}{N} \sum_{n=1}^N f(x; \xi_i) \tag{2}$$

where $\mathcal{D}$ is the data distribution and the second equality shows a special case of $\mathcal{D}$ be a uniform distribution over $N$ data points. We will focus on this finite sum setting in the subsequent discussion to be compatible with the finite-state Markov chain in Definition 3.1. Note that it is straightforward to extend our analysis to the general case of $\mathcal{D}$ if we extend our definition of Markov chain to the infinite state spaces. Next, we consider the following stochastic data echoing update:

$$x_{t+1} = x_t - \eta_t \nabla f(x_t; \mathcal{B}_t) \tag{3}$$

where $\eta_t$ is the learning rate and $\mathcal{B}_t$ denotes the mini-batch at step $t$. Different from that in the stochastic gradient descent where $\mathcal{B}_t$ is sampled independently for each step, we assume that $\mathcal{B}_{t+1}$ correlates with $\mathcal{B}_t$. More specifically, suppose we have $\mathcal{B}_t = \{z_{t,1}, ..., z_{t,B}\}$, where the size of the minibatch is $B$. Then we have the following sampling procedure:

$$z_{t+1,i} = \begin{cases} z_{t,i} & \text{with prob. } 1-p, \\ \xi_j & \text{with prob. } \frac{p}{N-1}, \ \forall \ \xi_j \neq z_{t,i}. \end{cases} \tag{4}$$

where $0 < p \leq 1$, and equation 4 shows $z_i$ is not updated with probability $1 - p$ and otherwise randomly load a new example from the dataset $\mathcal{D}$. Note that if we set $p = \frac{1}{M}$, then equation 3 is

equivalent to the classical data echoing update in equation 1 in expectation; if we set $p = \frac{N-1}{N}$, equation 3 degenerates to the standard stochastic gradient descent.

Following the sampling strategy in equation 4, we get $B$ independent Markov chains $\{z_i\}, i \in [B]$. The transition matrix $P$ of $\{z_i\}$ has diagonal elements of $1 - p$ and off-diagonal elements of $\frac{p}{N-1}$. We can identify two important properties from the transition matrix. Firstly, since $P$ is irreducible, aperiodic and reversible, the Markov chains $\{z_i\}, i \in [B]$ have a unique stationary distribution which is the uniform distribution, *i.e.* $\pi_{z_i} = \frac{1}{N}, i \in [B]$. By the definition of stationary distribution, we have $P(z_{t+\tau} \mid z_t) \to \pi$ as $\tau \to \infty$. In other words, for sufficiently large $\tau$, $z_{t+\tau}$ and $z_t$ are independent with each other, and this property plays an important role in our analysis.

Next, we derive the mixing time of the markov chain. We express the transition matrix $P$ as:

$$P = (1 - p) \times I + \frac{p}{N-1} \times (J - I)$$

where $J$ denotes the matrix whose elements are all ones. Following some standard linear algebra analysis, we can get $P$ has one eigenvalue of one and $N - 1$ eigenvalues of $(1 - p - \frac{p}{N-1})$ and following the properties of Markov chain in Sec. 3.1, we have:

$$\tau_{mix}(\epsilon) \le \lceil \lambda_P^{-1} \ln(\pi_{min}^{-1}\epsilon^{-1}) \rceil = \lceil \frac{N-1}{pN} \ln(N\epsilon^{-1}) \rceil = O\left(\frac{\ln(N\epsilon^{-1})}{p}\right)$$

In the case of $p = \frac{1}{M}$, *i.e.* the standard data echoing, we have the mixing time is $\tilde{O}(M)$. Therefore, a lower frequency of data loading leads to a larger mixing time. As the mixing time measures how fast the state distribution converges to the stationary distribution (uniform distribution in case of echoing), larger mixing time leads to a larger gradient estimation bias, thus slow down the convergence. Finally, Algorithm 1 presents a practical version of our stochastic data echoing algorithm, where we set $\{p_t\}$ to be a variable and depend on the training step. In the sequel, we will use the term 'data echoing' to refer to this stochastic version without special notation for simplicity.

## 3.3 THEORETICAL ANALYSIS OF ALGORITHM 1

In this section, we provide theoretical analysis to Algorithm 1. We first state some mild assumptions:

**Assumption 3.3** (Example Gradient Smoothness). *The function $f(x)$ in equation 2 is possibly non-convex, $f(x; \xi_i), i \in [N]$ is L-smooth i.e., we have:*

$$\|\nabla f(x; \xi_i) - \nabla f(y; \xi_i)\| \le L\|x - y\|,$$

*for all $x, y \in \mathbb{R}^d$ and $i \in [N]$.*

**Assumption 3.4.** *The function $f(x)$ is bounded from below,* i.e., *there exists $f^* = \inf_{x \in \mathcal{X}} f(x)$.*

**Assumption 3.5** (Bounded Gradient Dissimilarity). *The sample gradients have bounded dissimilarity to the full gradient,* i.e. *there exists a constant $\sigma$ such that:*

$$\|\nabla f(x; \xi^{(i)}) - \nabla f(x)\| \le \sigma, \forall\, i \in [N]$$

As stated in Assumption 3.3, we study the smooth non-convex optimization problems. Assumption 3.4 guarantees the minimization problem is well-defined. Finally, the bounded gradient dissimilarity assumption (Assumption 3.5) is weaker compared to the bounded gradient assumption made in (Agarwal et al., 2020). As the sample gradient can be unbounded while still satisfying Assumption 3.5.

A key step in our analysis is using the following shifted descent Lemma to bound one step progress of Algorithm 1 (See Lemma A.4 for proof):

**Lemma 3.6** (Shifted Descent Lemma). *Under Assumptions 3.3-3.5, for all $t \ge \tau \ge \tau_{mix}(\nu\pi_{min})$, $\nu < \frac{1}{4}$ and $\eta < \frac{1}{256L\tau}$, the one step progress of Algorithm 1 can be bounded as follows:*

$$\mathbb{E}[f(x_{t+1})] \le \mathbb{E}[f(x_t)] - \frac{\eta}{16}\mathbb{E}\left[\left\|\nabla f(x_{t-\tau})\right\|^2\right] + \eta\nu^2\sigma^2 + 128\eta^3\tau^2L^2\sigma^2 + \eta^2L\sigma^2$$

As shown by Lemma 3.6, the stochastic gradient $\nabla f(x_t; \mathcal{B}_t)$ evaluated over the mini-batch $\mathcal{B}_t$ leads to a good estimation of $\nabla f(x_{t-\tau})$. In fact, for sufficiently large $\tau$, $P(\mathcal{B}_t|x_{t-\tau})$ converges to the stationary distribution, which is the uniform data distribution in our case. As a result, we have $\mathbb{E}[\nabla f(x_{t-\tau}; \mathcal{B}_t)] \approx \nabla f(x_{t-\tau})$; then the bias of using $\nabla f(x_t; \mathcal{B}_t)$ to estimate $\nabla f(x_{t-\tau})$ is at the order of $\|\nabla f(x_{t-\tau}) - \nabla f(x_t)\|$ which can be bounded by variable drift $\|x_{t-\tau} - x_t\|$ by the smoothness assumption. See Lemma A.4 in the appendix for the detailed proof.

---

**Algorithm 2** Communication-Efficient Data-Echoing for Data Parallelism (Comm. Data-Echo)

---

1: **Input**: **Input**: $\{\eta_t\}$, learning rates; $\{p_t\}$, the probability of an example is replaced with a new one; $\{p_{t,}^{(c)}\}$, the probability of different nodes average local gradients; $K$, the number of nodes; $B$, the minibatch size. Initialize the state $x_1^{(k)} = x_1, k \in [K]$ for some state $x_1$.
2: **for** $t = 1$ to $T$ **do**
3:    **for** $k = 1$ to $K$ in parallel **do**
4:       Each node independently update the mini-batch $\mathcal{B}_t^{(k)}$ according to equation 4;
5:       Compute the stochastic gradient $d_t^{(k)} = \frac{K}{B} \sum_{z_t^{(k)} \in \mathcal{B}_t^{(k)}} \nabla f(x_t^{(k)}; z_t^{(k)})$ with $|\mathcal{B}_t^{(k)}| = \frac{B}{K}$
6:       $x_{t+1}^{(k)} = x_t^{(k)} - \eta_t d_t^{(k)}$
7:    **end for**
8:    Set $x_{t+1}^{(k)} = \frac{1}{K} \sum_{j=1}^{K} x_{t+1}^{(j)}$ with probability $p_t^{(c)}$
9: **end for**
10: **Return:** $x_a$ chosen uniformly randomly from $\{x_t\}_{t=1}^{T}$

---

*Remark* 3.7. Note that the type of shifted analysis is an existing technique in the Markov chain gradient descent literature (Sun et al., 2018; Even, 2023). However, our Lemma 3.6 leads to a tighter bound to the gradient estimation bias, where our bias term is $O(\eta\nu^2 + \eta^3\tau^2)$. In contrast, that state-of-art analysis by (Even, 2023) shows an bound of $O(\eta\nu^2 + \eta^2\tau)$ (See Section C.1 of (Even, 2023)). For small value $\eta$, our bound is tighter, so our Lemma 3.6 is useful to the analysis of general markov chain gradient descent.

Next, we are ready to show the convergence Theorem of Algorithm 1 (The proof is included in Theorem A.5):

**Theorem 3.8.** *Under Assumptions 3.3-3.5, we choose* $\eta = \min\left(\frac{1}{256L\tau}, \frac{1}{4}, \left(\frac{C_0 C_2}{T}\right)^{1/2}\right)$, *for some constants* $\tau = O(1/p)$, *then with any choice of minibatch sizes* $B \geq 1$, *the iterates generated from Algorithm 1 satisfy:*

$$\frac{1}{T} \sum_{t=1}^{T} \mathbb{E}\|\nabla f(x_t)\|^2 \leq \frac{C_0}{C_\eta T} + C_1 \left(\frac{C_0 C_2}{T}\right) + C_2 \left(\frac{C_0 C_2}{T}\right)^{1/2}$$

*where* $C_0 = 16(\Delta + \sigma^2/L)$, $C_1 = (16 + 16 * 128\tau^2 L^2)\sigma^2$, $C_2 = 16L\sigma^2$, $C_\eta = \min\left(\frac{1}{256L\tau}, \frac{1}{4}\right)$. $\Delta$ *denotes the initial sub-optimality.* $p$ *is the data loading probability; the stochasity comes from the randomness of the algorithm.*

Note that the last term in the above inequality dominates the convergence error for sufficiently large $T$. More precisely, we need $T \geq \max\left(\frac{C_1^2 C_0}{C_2}, \frac{C_0}{C_\eta^2 C_2^3}\right)$. Given this condition hold, to reach an $\epsilon$-stationary point, we need $T = O(C_0 C_2^3 \epsilon^{-2})$, then the number of data loading operations is $O(pBC_0 C_2^3 \epsilon^{-2})$, Note that $C_0$ and $C_2$ does not depend on $p$, so the number of data loading operations has linear speedup *w.r.t.* the data loading probability $p$. Recall that Agarwal et al. (2020) only showed that the number of data loading operations is $O(\epsilon^{-2})$ and does not benefit from reusing in-memory batches. Furthermore, the analysis of Agarwal et al. (2020) makes the bounded gradient assumption, while our analysis requires the weaker bounded gradient dissimilarity (Assumption 3.5) condition.

*Remark* 3.9. Note that we can not arbitrarily decrease $p$ to 0 in practice, as Theorem 3.8 requires a *burn-in stage*, *i.e.* we need $T \geq \max\left(\frac{C_1^2 C_0}{C_2}, \frac{C_0}{C_\eta^2 C_2^3}\right)$, recall that $C_\eta = O(\tau^{-1})$, $C_1 = O(\tau^2)$ and $\tau = O(p^{-1})$, so we need at least $T = O(p^{-4})$ to achieve the speedup effect *w.r.t.* $p$.

## 4   COMMUNICATION-EFFICIENT DATA ECHOING FOR DATA PARALLELISM

In this section, we propose a new communication-efficient data echoing for data parallelism. Suppose we have $K$ nodes for parallel computation. In data parallelism, at each training step, the mini-batch $\mathcal{B}_t$ is divided into $K$ subsets $\{\mathcal{B}_t^{(k)}, k \in [K]\}$, and each node evaluates stochastic gradient over a different subset, *i.e.* evaluate $\nabla f(x_t; \mathcal{B}_t^{(k)})$. Then a gradient average operation is performed across

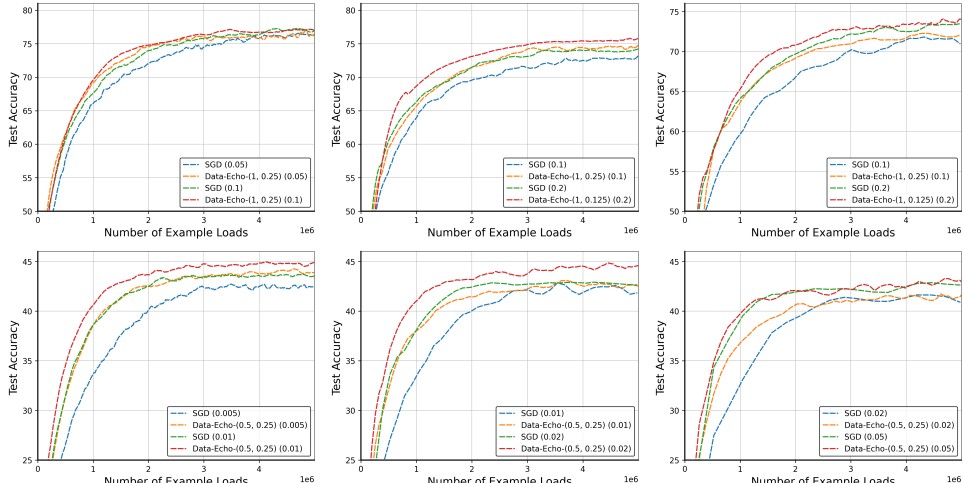

Figure 2: **Comparison between Data Echoing with Diminishing Load Probability *vs* SGD**. The learning rate for each algorithm is specified in parentheses, *e.g.*, SGD (0.1) denotes a learning rate of 0.1. The first row presents results for CIFAR-10 trained on MobileNet-V2, while the second row shows results for CIFAR-100 trained on ResNet18. Batch sizes of 128, 256, and 512 are shown in the first, second, and third columns, respectively.

all nodes. The update step of data parallelism is as follows:

$$x_{t+1} = x_t - \frac{\eta_t}{K} \sum_{k=1}^{K} \nabla f(x_t; \mathcal{B}_t^{(k)}) \tag{5}$$

Note that the vanilla data echoing algorithm only reduces the number of data loading operations (as shown by Theorem 3.8), but the number of training steps $T$ is not changed. In the data parallelism setting, this means the communication cost across the nodes is not reduced, which undermines the significance of doing data echoing. In fact, one way to mitigate this challenge is by reducing the frequency of gradient average. More specifically, we perform the average operation with probability $p^{(c)}$. Recall that the data loading operation is performed with probability $p$. By setting $p^{(c)} \leq p$, the communication overhead between compute nodes can be eliminated.. In Algorithm 2, we formalize our communication-efficient data echoing algorithm.

### 4.1 THEORETICAL ANALYSIS OF ALGORITHM 2

In this section, we study the convergence property of Algorithm 2. Note that the bias of gradient estimation is increased if we reduce the frequency of gradient average, thus slows down the convergence rate. However, we show its effect to convergence is bounded as long as the interval between two gradient average operations are bounded. More specifically, we make the following assumption:

**Assumption 4.1** (Bounded Gradient Average Interval). *The interval between two gradient average steps are upper-bounded by $I$.*

Then we are ready to show the convergence Theorem of Algorithm 2 (See Theorem A.3 for proof):

**Theorem 4.2.** *Under Assumptions 3.3-3.5 and Assumption 4.1, we choose $\eta = \min\left(\frac{1}{256L\tau}, \frac{1}{4LI}, \frac{1}{4c_\nu}, \left(\frac{C_0 C_2}{T}\right)^{1/2}\right)$, for some constants $\tau = O(1/p)$, then with any choice of mini-batch sizes $B \geq 1$, and number of local updates, $I \geq 1$, the iterates generated from Algorithm 2 satisfy:*

$$\frac{1}{T} \sum_{t=1}^{T} \mathbb{E}\|\nabla f(\bar{x}_t)\|^2 \leq \frac{C_0}{C_\eta T} + C_1 \left(\frac{C_0 C_2}{T}\right) + C_2 \left(\frac{C_0 C_2}{T}\right)^{1/2}$$

*where $C_0 = 16(\Delta + \sigma^2/L)$, $C_1 = (32 + 16 * 128\tau^2 L^2 + 256L^2(I-1)^2\tau)\sigma^2$, $C_2 = 16L\sigma^2$, $C_\eta = \min\left(\frac{1}{256L\tau}, \frac{1}{4LI}, \frac{1}{4}\right)$. $\Delta$ denotes the initial sub-optimality. The stochasity comes from the randomness of the algorithm.*

The error bound in Theorem 4.2 resembles that in Theorem 3.8, except the additional term $256(I - 1)^2 L^2 \tau$ in the constant $C_1$ and the additional regularity to the learning rate where $\eta < 1/(4LI)$. Note that the term $256(I - 1)^2 L^2 \tau$ bounds the drift of local variables $\{x_t^{(k)}, k \in [K]\}$ (See Lemma A.1 in the appendix for more details).

Similar to Theorem 3.8, to reach an $\epsilon$-stationary point, Theorem 4.2 needs $T = O(\epsilon^{-2})$ (we omit the constants $C_0$ and $C_2$) and the number of data loading operations is $O(pB\epsilon^{-2})$. Thus Algorithm 2 also has linear speedup *w.r.t.* $p$. Furthermore, the number of gradient average operations are $O(p^{(c)}B\epsilon^{-2})$. Note that Theorem 4.2 allows the $I$ grows at the order of $O(T^{1/4})$ without affecting the dominant term ($C_1$ only affects the second term of the error bound). So we can set $p^{(c)} = \frac{1}{I} \le p$ such that the communication across compute node is comparable to data loading operations.

## 5 NUMERICAL EXPERIMENTS

In this section, we perform experiments to verify the efficacy of our proposed algorithms through both image classification tasks and language modeling tasks. More specifically, for the image classification task, we use CIFAR-10 (Krizhevsky et al., 2009) and CIFAR-100 (Krizhevsky et al., 2009) and choose ResNets (He et al., 2016) and MobileNet-V2 (Sandler et al., 2018) models. For the language modeling task, we use WikiText-2 (Merity et al., 2016) dataset over the GPT-2 model (Radford et al., 2019). All experimental results are averaged over 5 independent runs and we report the average value. Next, in Section 5.1, we consider the single node setting and focus on the effect of data loading probability schedule $\{p_t\}, t \in [T]$. Then in Section 5.2, we study the multi-node data parallelism setting, where we test our communication-efficient data echoing algorithm. All experiments are run on a machine with an Intel Xeon Gold 6248 CPU and 4 Nvidia Tesla A6000 GPUs. The code is written in Pytorch. We simulate the communication among different GPUs through the Pytorch.distributed package (Li et al., 2020).

### 5.1 RESULTS UNDER SINGLE NODE SETTING

In this section, we examine how the data loading probability $p$ impacts model training performance. When using a constant $p = \frac{1}{M}$ in Algorithm 1, the expected behavior aligns with classical data echoing with $M$ repeat steps, and we use this constant schedule as the starting point of our experiments. We explore different values of $p$ across various training settings, including model architecture, datasets, and learning rates. Following the experimental setup in Choi et al. (2019), we measure training cost by the number of example loads, as this metric is more reproducible across different hardware compared to wall-clock time (Choi et al., 2019).

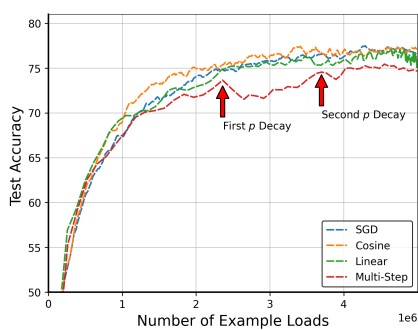

Figure 3: **Comparison among different Loading Probability schedule**. We train a MobileNet-V2 model on the CIFAR-10 dataset using a learning rate of 0.1 and a batch size of 128.

As shown in Figure 4, the Data-Echoing algorithm (for moderately large $p$) converges slower than the standard SGD algorithm. In fact, when the batchsize $B$ is much smaller compared to the dataset size (*e.g.* 128 compared to 50000 in Figure 4), reusing the same batch of data during training can lead to overfitting to that specific batch and thus slow down the overall convergence rate. Furthermore, we observe that the test accuracy converges to a similar point across different $p$ values, with smaller $p$ values causing slower initial convergence but faster convergence in the later stages of training. For example, in the third plot of Figure 4, Data-Echo with $p = 0.25$ converges slower than $p = 0.5$ in the early training stage, but it catches up and finally converges to a slightly better accuracy. This observation is consistent to our theoretical analysis. In Theorem 3.8, we show Algorithm 1 has a 'burn-in stage' which is at the order of $O(p^{-4})$. Naturally, smaller values of $p$ requires more training steps to show its acceleration effects.

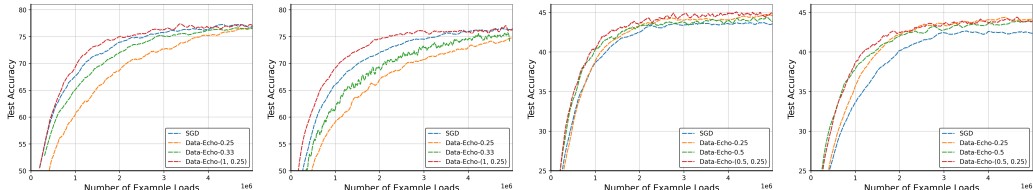

Figure 4: **Comparison between Diminishing *vs* Constant Loading Probability**. The left two figures show results for CIFAR-10 (trained with MobileNet-V2) using learning rates of 0.05 and 0.1, respectively. The right two figures present results for CIFAR-100 (trained with ResNet-18) using learning rates of 0.005 and 0.01, respectively. A batch size of 128 is used in all cases.

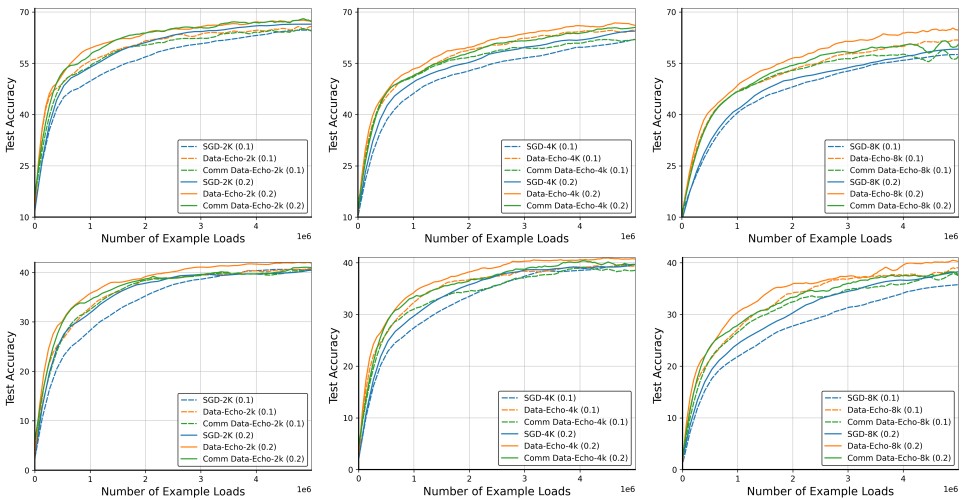

Figure 5: **Comparison between Communication-Efficient Data Echoing *vs* SGD**. Data-Echo and Comm. Data-Echo use the same $p$ schedule, with $p^{(c)}$ set equal to $p$, ensuring that the communication cost for Comm. Data-Echo matches that of vanilla SGD. The top row presents results for CIFAR-10 trained on MobileNet-V2, while the bottom row shows results for CIFAR-100 trained on ResNet-18.

To mitigate this challenge, we propose a diminishing schedule for the data loading probability $p$. More specifically, we adopt the following cosine schedule:

$$p_E = p_{min} + (p_{max} - p_{min})cos\left(\frac{E\pi}{2E_{max}}\right) \tag{6}$$

where $E$ ($E_{max}$) are the number of epochs (max epochs) and $0 < p_{min} < p_{max} < 1$. We use *Data-Echo ($p_{max}$, $p_{min}$)* to denote the data echoing using this cosine schedule. As shown by the results in Figure 4, data-echoing using the cosine schedule outperforms that using constant data loading probability $p$.

Note that it is also possible to use other diminishing schedules for the data loading probability $p$. As shown in Figure 3, we compare the cosine schedule with two more types of schedules: Linear Decay and Multi-Step Decay. For Linear decay, $p$ is decreased from $p_{max}$ to $p_{min}$ linearly throughout training, while for multi-step decay, we decrease the value of $p$ at $0.5E_{max}$ and $0.75E_{max}$ by half, respectively. In Figure 3, the multi-step accuracy curve has a big drop when we decrease the $p$, which could be caused by the sudden change of input data distribution, and this observation motivates us to use a smoother decaying schedule for $p$. As for linear decay schedule, it is slightly worse than the cosine schedule. Note that this suggests that the linear decay schedule might decrease the value of $p$ too fast compared to the cosine schedule. In the remainder of experiments, we will focus on the cosine schedule. Finally, in Figure 2 (the image classification tasks) and Figure 7 (the language modeling task), we verify the effectiveness of the cosine schedule under different settings and we observe that the data echoing with cosine decay schedule consistently outperforms SGD.

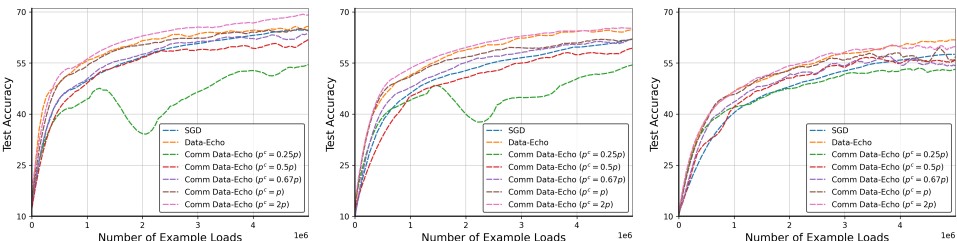

Figure 6: **Ablation study of the model average probability** $p^{(c)}$. We vary the ratio between $p^{(c)}$ and $p$. The results are for the MobileNet-V2 trained on the CIFAR-10 dataset with learning rate 0.1. We use batchsize 2048, 4096 and 8192 from left to right.

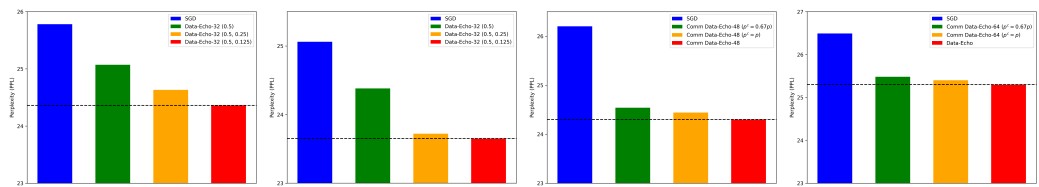

Figure 7: **Numerical Results for the Language Modeling Task.** The two figures on the left illustrate results for the single-node setting with learning rates of $5 \times 10^{-5}$ and $10^{-4}$, respectively. The curves are labeled in the format '*method-batchsize (p schedule)*'. The two figures on the right correspond to the multi-node setting with batch sizes of 48 and 64, respectively, where the curve labels follow the format '*method-batch size ($p^{(c)}$ schedule)*'. All results are based on training for 30 epochs.

### 5.2 Results under Data Parallelism Setting

In this section, we show that our *Comm. Data-Echo* (Algorithm 2) can effectively reduce communication cost of data echoing in data parallelism while still keeping its benefit in reducing the number of data loading operations. We run experiments with 4 nodes and evenly divide the data to each node at each step. Firstly, in Figure 5, we set $p^{(c)} = p$, *i.e.* we set the probability of performing data loading the same as performing gradient average at each step. As a result, the amount of communication among GPUs needed for data echoing and SGD are the same (in expectation) given same number of data loading operations. Then we observe in Figure 5 that the *Comm. Data Echo* gets comparable performance as the vanilla data echoing while outperform the vanilla SGD, which effectively verify the effectiveness of *Comm. Data Echo*. Next, in Figure 6, we further study how the choice $p^{(c)}$ affects the training performance, where we vary the value of $p^{(c)}$ in the range of $(0.25p, 2p)$. As shown in the figure, we can set $p^{(c)}$ as low as $0.67p$ while still outperforming the vanilla SGD. Finally, the results shown in the right two figures of Figure 7 also verify the effectiveness of our *Comm. Data Echo* in reducing the communication overhead of data echoing.

## 6 Conclusion

In this work, we explore the properties of the data echoing technique. First, we provide a tighter analysis of data echoing, demonstrating that its stochastic formulation achieves linear speedup with respect to data loading probability. Next, we examine the application of data echoing in the context of data parallelism and propose a novel algorithm that reduces the frequency of model averaging, thereby improving communication efficiency. Theoretically, we show that our communication-efficient data echoing algorithm lowers the communication cost in data parallelism while preserving the benefit of reduced data loading operations. Finally, we empirically validate the effectiveness of data echoing and introduce a cosine diminishing schedule for data loading probability. We also conduct extensive numerical experiments to confirm the efficacy of our communication-efficient algorithm in the data parallelism setting.

## REPRODUCIBILITY STATEMENT

For theoretical results, we state all assumptions in Section 3.3 and Section 4.1, and the detailed proof of all lemmas and theorems is included in the appendix. For numerical results, we describe the details including datasets, models, and various hyper-parameters choices.

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

## A  Proof for Convergence Theorem

In this section, we provide proof to our convergence theorem Theorem 3.8 and Theorem 4.2. Since the single node setting (Theorem 3.8) can be viewed as a special case of the data-parallelism setting (Theorem 3.8), we first prove Theorem 4.2 in Section A.1 and then provide the proof for Theorem 4.2 in Section A.2 for completeness.

### A.1  Proof of Convergence Theorem under Data-Parallelism Setting

We define $\bar{t}_s$ be the time step when nodes average model states, $\bar{x}_t = \frac{1}{K} \sum_{k=1}^{K} x_t^{(k)}$, $\bar{d}_t = \frac{1}{K} \sum_{k=1}^{K} d_t^{(k)}$. We define the filtration $\mathcal{F}_t$ as the sigma algebra generated by iterates $x_1^{(k)}, x_2^{(k)}, \ldots, x_t^{(k)}$ as

$$\mathcal{F}_t = \sigma(x_1^{(k)}, x_2^{(k)}, \ldots, x_t^{(k)}, \text{ for all } k \in [K]).$$

Next, we bound the error accumulated via the iterates generated by the local updates of Algorithm 2.

**Lemma A.1** (Local State Drift). *Under Assumption 4.1, then for the choice of stepsize $\eta \leq \frac{1}{4LI}$, the iterates $x_t^{(k)}$ for each $k \in [K]$ generated from Algorithm 2 satisfy:*

$$\sum_{t=1}^{T} \frac{1}{K} \sum_{k=1}^{K} \mathbb{E}\|x_t^{(k)} - \bar{x}_t\|^2 \leq 4\eta^2(I-1)^2\sigma^2 T$$

*where the expectation is w.r.t the stochasticity of the algorithm.*

*Proof.* Note that at $t = \bar{t}_{s-1}$ with $s \in [S]$, $x_t^{(k)} = \bar{x}_t$, for all $k$, therefore, we have $\frac{1}{K} \sum_{k=1}^{K} \|x_{\bar{t}_{s-1}}^{(k)} - \bar{x}_{\bar{t}_{s-1}}\|^2 = 0$. Moreover, for $t \in [\bar{t}_{s-1}+1, \bar{t}_s - 1]$, with $s \in [S]$, we have:

$$\frac{1}{K} \sum_{k=1}^{K} \|x_t^{(k)} - \bar{x}_t\|^2 = \frac{1}{K} \sum_{k=1}^{K} \left\| x_{\bar{t}_{s-1}}^{(k)} - \bar{x}_{\bar{t}_{s-1}} - \left( \sum_{\ell=\bar{t}_{s-1}}^{t-1} \eta d_\ell^{(k)} - \sum_{\ell=\bar{t}_{s-1}}^{t-1} \eta \bar{d}_\ell \right) \right\|^2$$

$$= \frac{\eta^2}{K} \sum_{k=1}^{K} \left\| \sum_{\ell=\bar{t}_{s-1}}^{t-1} (d_\ell^{(k)} - \bar{d}_\ell) \right\|^2$$

$$\overset{(a)}{\leq} \frac{2\eta^2}{K} \sum_{k=1}^{K} \left\| \sum_{\ell=\bar{t}_{s-1}}^{t-1} [d_\ell^{(k)} - \nabla f(x_\ell^{(k)}) - \frac{1}{K} \sum_{j=1}^{K} (d_\ell^{(j)} - \nabla f(x_\ell^{(j)}))] \right\|^2$$

$$+ \frac{2\eta^2}{K} \sum_{k=1}^{K} \left\| \sum_{\ell=\bar{t}_{s-1}}^{t-1} \left( \nabla f(x_\ell^{(k)}) - \frac{1}{K} \sum_{j=1}^{K} \nabla f(x_\ell^{(j)}) \right) \right\|^2$$

$$\overset{(b)}{\leq} \frac{2\eta^2}{K} \sum_{k=1}^{K} \left\| \sum_{\ell=\bar{t}_{s-1}}^{t-1} (d_\ell^{(k)} - \nabla f(x_\ell^{(k)})) \right\|^2$$

$$+ \frac{2\eta^2}{K} \sum_{k=1}^{K} \left\| \sum_{\ell=\bar{t}_{s-1}}^{t-1} \left( \nabla f(x_\ell^{(k)}) - \frac{1}{K} \sum_{j=1}^{K} \nabla f(x_\ell^{(j)}) \right) \right\|^2 \quad (7)$$

where $(a)$ is by the generalized triangle inequality; $(b)$ is based on the property of separating mean and variance.

Taking expectation on both sides and let us next consider each term of equation 7 above separately, we have for any $k \in [K]$ from the first term of equation 7 above

$$\mathbb{E}\left\| \sum_{\ell=\bar{t}_{s-1}}^{t-1} (d_\ell^{(k)} - \nabla f(x_\ell^{(k)})) \right\|^2 \overset{(a)}{=} \sum_{\ell=\bar{t}_{s-1}}^{t-1} \mathbb{E}\left\| d_\ell^{(k)} - \nabla f(x_\ell^{(k)}) \right\|^2 \leq (I-1)\sigma^2 \quad (8)$$

where we use the independent sampling property and bounded gradient bias assumption.

Next, we consider the second term of equation 7 for any $k \in [K]$, we have:

$$\sum_{k=1}^{K} \mathbb{E} \left\| \sum_{\ell=\bar{t}_{s-1}}^{t-1} \left( \nabla f(x_\ell^{(k)}) - \frac{1}{K} \sum_{j=1}^{K} \nabla f(x_\ell^{(j)}) \right) \right\|^2$$

$$\leq (I-1) \sum_{\ell=\bar{t}_{s-1}}^{t-1} \sum_{k=1}^{K} \mathbb{E} \left\| \nabla f(x_\ell^{(k)}) - \frac{1}{K} \sum_{j=1}^{K} \nabla f(x_\ell^{(j)}) \right\|^2$$

$$\leq (I-1) \sum_{\ell=\bar{t}_{s-1}}^{t-1} \left[ 2 \sum_{k=1}^{K} \mathbb{E} \left\| \nabla f(x_\ell^{(k)}) - \nabla f(\bar{x}_\ell) \right\|^2 + 2 \sum_{k=1}^{K} \mathbb{E} \left\| \frac{1}{K} \sum_{j=1}^{K} (\nabla f(\bar{x}_\ell) - \nabla f(x_\ell^{(j)})) \right\|^2 \right]$$

$$\leq 4L^2(I-1) \sum_{\ell=\bar{t}_{s-1}}^{t-1} \sum_{k=1}^{K} \mathbb{E} \left\| x_\ell^{(k)} - \bar{x}_\ell \right\|^2 \tag{9}$$

where we use the generalized triangle inequality and the bounded gradient bias assumption.

Substituting equation 8 and equation 9 in equation 7 and taking expectation on both sides we get:

$$\frac{1}{K} \sum_{k=1}^{K} \mathbb{E} \| x_t^{(k)} - \bar{x}_t \|^2 \leq 2\eta^2 (I-1)^2 \sigma^2 + 8\eta^2 L^2 (I-1) \sum_{\ell=\bar{t}_{s-1}}^{t-1} \frac{1}{K} \sum_{k=1}^{K} \mathbb{E} \| x_\ell^{(k)} - \bar{x}_\ell \|^2$$

Summing both sides from $t = \bar{t}_{s-1}$ to $\bar{t}_s - 1$, we get:

$$\sum_{t=\bar{t}_{s-1}}^{\bar{t}_s-1} \frac{1}{K} \sum_{k=1}^{K} \mathbb{E} \| x_t^{(k)} - \bar{x}_t \|^2 \leq 2\eta^2 (I-1)^2 \sigma^2 I + 8L^2(I-1)\eta^2 \sum_{t=\bar{t}_{s-1}}^{\bar{t}_s-1} \sum_{\ell=\bar{t}_{s-1}}^{t-1} \frac{1}{K} \sum_{k=1}^{K} \mathbb{E} \| x_\ell^{(k)} - \bar{x}_\ell \|^2$$

$$\overset{(a)}{\leq} 2\eta^2 (I-1)^2 \sigma^2 I + 8L^2(I-1)\eta^2 I \sum_{t=\bar{t}_{s-1}}^{\bar{t}_s-1} \frac{1}{K} \sum_{k=1}^{K} \mathbb{E} \| x_t^{(k)} - \bar{x}_t \|^2$$

where $(a)$ uses that fact that $t \leq \bar{t}_s - 1$ and $t_s - t_{s-1} \leq I$ for all $s \in [S]$.

Finally, summing over $s \in [S]$ and using $T = SI$ we get:

$$\sum_{t=1}^{T} \frac{1}{K} \sum_{k=1}^{K} \mathbb{E} \| x_t^{(k)} - \bar{x}_t \|^2 \leq 2\eta^2 (I-1)^2 \sigma^2 T + 8L^2 I^2 \eta^2 \sum_{t=1}^{T} \frac{1}{K} \sum_{k=1}^{K} \mathbb{E} \| x_t^{(k)} - \bar{x}_t \|^2$$

Rearranging the terms, we get

$$(1 - 8L^2 I^2 \eta^2) \sum_{t=1}^{T} \frac{1}{K} \sum_{k=1}^{K} \mathbb{E} \| x_t^{(k)} - \bar{x}_t \|^2 \leq 2\eta^2 (I-1)^2 \sigma^2 T$$

Finally, using the fact that $\eta \leq \frac{1}{4LI}$ we have $1 - 8L^2 I^2 \eta^2 \geq 1/2$. Multiplying, both sides by 2 we get

$$\sum_{t=1}^{T} \frac{1}{K} \sum_{k=1}^{K} \mathbb{E} \| x_t^{(k)} - \bar{x}_t \|^2 \leq 4\eta^2 (I-1)^2 \sigma^2 T$$

Therefore, the lemma is proved. □

**Lemma A.2** (Shifted Descent Lemma). *For all $t \geq \tau \geq \tau_{mix}(\nu \pi_{min})$, $\nu < \frac{1}{4}$ and $\eta < \frac{1}{256L\tau}$, the one step progress can be bounded as follows:*

$$\mathbb{E}[f(\bar{x}_{t+1})] \leq \mathbb{E} \left[ f(\bar{x}_t) - \frac{\eta}{16} \mathbb{E} \left[ \left\| \nabla f(\bar{x}_{t-\tau}) \right\|^2 \right] + 2\eta\nu^2\sigma^2 + 128\tau^2\eta^3 L^2\sigma^2 + \eta^2 L\sigma^2 \right.$$

$$+ 4\eta^2 L^3 \frac{1}{K} \sum_{k=1}^{K} \| x_t^{(k)} - \bar{x}_t \|^2 + \frac{\eta L^2}{K} \sum_{k=1}^{K} \left\| x_{t-\tau}^{(k)} - \bar{x}_{t-\tau} \right\|^2$$

$$\left. + 128\tau\eta^3 L^4 \sum_{\ell=max(t-\tau,0)}^{t-1} \frac{1}{K} \sum_{k=1}^{K} \left\| x_\ell^{(k)} - \bar{x}_\ell \right\|^2 \right]$$

*Proof.* By the smoothness of $f$, we have:

$$\mathbb{E}[f(\bar{x}_{t+1})] \leq \mathbb{E}\Big[f(\bar{x}_t) - \eta\langle\nabla f(\bar{x}_t), \bar{d}_t\rangle + \frac{\eta^2 L}{2}\|\bar{d}_t\|^2\Big]$$

$$= \mathbb{E}\Big[f(\bar{x}_t) - \eta\Big\langle\nabla f(\bar{x}_t), \frac{1}{K}\sum_{k=1}^{K}\nabla f(x_t^{(k)}; \mathcal{B}_t^{(k)})\Big\rangle + \frac{\eta^2 L}{2}\|\bar{d}_t\|^2\Big] \qquad (10)$$

For the second term in the above inequality, we split it into three terms:

$$\Big\langle\nabla f(\bar{x}_t), \frac{1}{K}\sum_{k=1}^{K}\nabla f(x_t^{(k)}; \mathcal{B}_t^{(k)})\Big\rangle = \Big\langle\nabla f(\bar{x}_{t-\tau}), \frac{1}{K}\sum_{k=1}^{K}\nabla f(x_{t-\tau}^{(k)}; \mathcal{B}_t^{(k)})\Big\rangle$$

$$+ \Big\langle\nabla f(\bar{x}_t) - \nabla f(\bar{x}_{t-\tau}), \frac{1}{K}\sum_{k=1}^{K}\nabla f(x_{t-\tau}^{(k)}; \mathcal{B}_t^{(k)})\Big\rangle$$

$$+ \Big\langle\nabla f(\bar{x}_t), \frac{1}{K}\sum_{k=1}^{K}(\nabla f(x_t^{(k)}; \mathcal{B}_t^{(k)}) - \nabla f(x_{t-\tau}^{(k)}; \mathcal{B}_t^{(k)}))\Big\rangle$$

$$(11)$$

For the first term in equation 11, we have:

$$\mathbb{E}\Big[\Big\langle\nabla f(\bar{x}_{t-\tau}), \frac{1}{K}\sum_{k=1}^{K}\nabla f(x_{t-\tau}^{(k)}; \mathcal{B}_t^{(k)})\Big\rangle\Big] = \mathbb{E}_{\mathcal{F}_{t-\tau}}\Big[\Big\langle\nabla f(\bar{x}_{t-\tau}), \frac{1}{K}\sum_{k=1}^{K}\mathbb{E}[\nabla f(x_{t-\tau}^{(k)}; \mathcal{B}_t^{(k)})]\Big\rangle\Big]$$

$$= \frac{1}{2}\mathbb{E}_{\mathcal{F}_{t-\tau}}\Big[\Big\|\nabla f(\bar{x}_{t-\tau})\Big\|^2 + \Big\|\frac{1}{K}\sum_{k=1}^{K}\mathbb{E}[\nabla f(x_{t-\tau}^{(k)}; \mathcal{B}_t^{(k)})]\Big\|^2 - \Big\|\nabla f(\bar{x}_{t-\tau}) - \frac{1}{K}\sum_{k=1}^{K}\mathbb{E}[\nabla f(x_{t-\tau}^{(k)}; \mathcal{B}_t^{(k)})]\Big\|^2\Big]$$

$$\geq \frac{1}{2}\mathbb{E}_{\mathcal{F}_{t-\tau}}\Big[\Big\|\nabla f(\bar{x}_{t-\tau})\Big\|^2 + \Big\|\frac{1}{K}\sum_{k=1}^{K}\mathbb{E}[\nabla f(x_{t-\tau}^{(k)}; \mathcal{B}_t^{(k)})]\Big\|^2 - 2\Big\|\nabla f(\bar{x}_{t-\tau}) - \frac{1}{K}\sum_{k=1}^{K}\mathbb{E}[\nabla f(\bar{x}_{t-\tau}; \mathcal{B}_t^{(k)})]\Big\|^2$$

$$- 2\Big\|\frac{1}{K}\sum_{k=1}^{K}\mathbb{E}[\nabla f(x_{t-\tau}^{(k)}; \mathcal{B}_t^{(k)})] - \frac{1}{K}\sum_{k=1}^{K}\mathbb{E}[\nabla f(\bar{x}_{t-\tau}; \mathcal{B}_t^{(k)})]\Big\|^2\Big]$$

$$\geq \frac{1}{2}\mathbb{E}_{\mathcal{F}_{t-\tau}}\Big[\Big\|\nabla f(\bar{x}_{t-\tau})\Big\|^2 + \Big\|\frac{1}{K}\sum_{k=1}^{K}\mathbb{E}[\nabla f(x_{t-\tau}^{(k)}; \mathcal{B}_t^{(k)})]\Big\|^2 - 2\Big\|\nabla f(\bar{x}_{t-\tau}) - \frac{1}{K}\sum_{k=1}^{K}\mathbb{E}[\nabla f(\bar{x}_{t-\tau}; \mathcal{B}_t^{(k)})]\Big\|^2$$

$$- \frac{2L^2}{K}\sum_{k=1}^{K}\Big\|x_{t-\tau}^{(k)} - \bar{x}_{t-\tau}\Big\|^2\Big] \qquad (12)$$

Then we use the property that the distribution of $\mathcal{B}_t$ is close to the stationary distribution conditioned on $\mathcal{F}_{t-\tau}$, i.e. $|\mathbb{P}(z_{t,i} = j) - \pi_j| \leq \nu\pi_j$ when $t \geq \tau \geq \tau_{mix}(\nu\pi_{min})$, then for the difference term in the above inequality, we have:

$$\mathbb{E}_{\mathcal{F}_{t-\tau}}\Big[\Big\|\nabla f(\bar{x}_{t-\tau}) - \mathbb{E}[\nabla f(\bar{x}_{t-\tau}; \mathcal{B}_t)]\Big\|^2\Big]$$

$$\leq \mathbb{E}_{\mathcal{F}_{t-\tau}}\Big[\frac{1}{B}\sum_{z_{t,i}, i\in[B]}\Big\|\sum_{j\in[N]}(\mathbb{P}(z_{t,i} = j) - \pi_j)\nabla f(\bar{x}_{t-\tau}, j)\Big\|^2\Big]$$

$$\leq \mathbb{E}_{\mathcal{F}_{t-\tau}}\Big[\frac{1}{B}\sum_{z_{t,i}, i\in[B]}\nu^2\sum_{j\in[N]}\pi_j\Big\|\nabla f(\bar{x}_{t-\tau}, j)\Big\|^2\Big] \overset{(a)}{\leq} \mathbb{E}_{\mathcal{F}_{t-\tau}}\Big[2\nu^2\Big\|\nabla f(\bar{x}_{t-\tau})\Big\|^2 + 2\nu^2\sigma^2\Big]$$

where the first inequality is by the generalized triangle inequality and the second inequality is by the Jensen inequality over $L_2$ norm, while in (a), we use the bounded bias assumption, substitute this

back to equation 12 and use the condition that $\nu < \frac{1}{4}$, to have:

$$\mathbb{E}\left[\left\langle \nabla f(\bar{x}_{t-\tau}), \frac{1}{K}\sum_{k=1}^{K}\nabla f(x_{t-\tau}^{(k)};\mathcal{B}_t^{(k)})\right\rangle\right]$$

$$\geq \frac{1}{2}\mathbb{E}_{\mathcal{F}_{t-\tau}}\left[\frac{1}{2}\left\|\nabla f(\bar{x}_{t-\tau})\right\|^2 + \left\|\frac{1}{K}\sum_{k=1}^{K}\mathbb{E}[\nabla f(x_{t-\tau}^{(k)};\mathcal{B}_t^{(k)})]\right\|^2 - \frac{2L^2}{K}\sum_{k=1}^{K}\left\|x_{t-\tau}^{(k)} - \bar{x}_{t-\tau}\right\|^2 - 4\nu^2\sigma^2\right]$$

$$\geq \frac{1}{4}\mathbb{E}\left[\left\|\nabla f(\bar{x}_{t-\tau})\right\|^2\right] + \frac{1}{2}\left\|\frac{1}{K}\sum_{k=1}^{K}\mathbb{E}[\nabla f(x_{t-\tau}^{(k)};\mathcal{B}_t^{(k)})]\right\|^2 - \frac{L^2}{K}\sum_{k=1}^{K}\mathbb{E}\left[\left\|x_{t-\tau}^{(k)} - \bar{x}_{t-\tau}\right\|^2\right] - 2\nu^2\sigma^2$$

$$\tag{13}$$

Next for the second term of equation 11, we have:

$$\left\langle \nabla f(\bar{x}_t) - \nabla f(\bar{x}_{t-\tau}), \frac{1}{K}\sum_{k=1}^{K}\nabla f(x_{t-\tau}^{(k)};\mathcal{B}_t^{(k)})\right\rangle \geq -\left\|\nabla f(\bar{x}_t) - \nabla f(\bar{x}_{t-\tau})\right\| \times \left\|\frac{1}{K}\sum_{k=1}^{K}\nabla f(x_{t-\tau}^{(k)};\mathcal{B}_t^{(k)})\right\|$$

$$\geq -\frac{1}{2}(\tau\eta^2L^2\sum_{\ell=max(t-\tau,0)}^{t-1}\left\|\bar{d}_\ell\right\|^2 + \left\|\frac{1}{K}\sum_{k=1}^{K}\nabla f(x_{t-\tau}^{(k)};\mathcal{B}_t^{(k)})\right\|^2)$$

$$\geq -\frac{1}{2}(2\tau\eta^2L^2\sum_{\ell=max(t-\tau,0)}^{t-1}\left\|\frac{1}{K}\sum_{k=1}^{K}\nabla f(x_\ell^{(k)})\right\|^2 + 2\tau^2\eta^2L^2\sigma^2 + \left\|\frac{1}{K}\sum_{k=1}^{K}\nabla f(x_{t-\tau}^{(k)};\mathcal{B}_t^{(k)})\right\|^2)$$

$$\geq -\frac{1}{2}\Bigg(4\tau\eta^2L^2\sum_{\ell=max(t-\tau,0)}^{t-1}\left\|\nabla f(\bar{x}_\ell)\right\|^2 + 2\tau^2\eta^2L^2\sigma^2$$

$$+ 4\tau\eta^2L^4\sum_{\ell=max(t-\tau,0)}^{t-1}\frac{1}{K}\sum_{k=1}^{K}\left\|x_\ell^{(k)} - \bar{x}_\ell\right\|^2 + \left\|\frac{1}{K}\sum_{k=1}^{K}\nabla f(x_{t-\tau}^{(k)};\mathcal{B}_t^{(k)})\right\|^2\Bigg)$$

and for the third term in equation 11:

$$\left\langle \nabla f(\bar{x}_t), \frac{1}{K}\sum_{k=1}^{K}\left(\nabla f(x_t^{(k)};\mathcal{B}_t^{(k)}) - \nabla f(x_{t-\tau}^{(k)};\mathcal{B}_t^{(k)})\right)\right\rangle$$

$$\geq -\frac{1}{K}\sum_{k=1}^{K}\left\|\sum_{\ell=max(t-\tau,0)}^{t-1}\eta L d_\ell^{(k)}\right\| \times \left\|\nabla f(\bar{x}_t)\right\|$$

$$\geq -\frac{1}{K}\sum_{k=1}^{K}(16\tau\eta^2L^2\sum_{\ell=max(t-\tau,0)}^{t-1}\left\|d_\ell^{(k)}\right\|^2 + \frac{1}{32}\left\|\nabla f(\bar{x}_t)\right\|^2)$$

$$\geq -\frac{1}{K}\sum_{k=1}^{K}(32\tau\eta^2L^2\sum_{\ell=max(t-\tau,0)}^{t-1}\left\|\nabla f(x_\ell^{(k)})\right\|^2 + \frac{1}{32}\left\|\nabla f(\bar{x}_t)\right\|^2 + 32\tau^2\eta^2L^2\sigma^2)$$

$$\geq -(64\tau\eta^2L^4\sum_{\ell=max(t-\tau,0)}^{t-1}\frac{1}{K}\sum_{k=1}^{K}\left\|x_\ell^{(k)} - \bar{x}_\ell\right\|^2$$

$$+ 64\tau\eta^2L^2\sum_{\ell=max(t-\tau,0)}^{t-1}\left\|\nabla f(\bar{x}_\ell)\right\|^2 + \frac{1}{32}\left\|\nabla f(\bar{x}_t)\right\|^2 + 32\tau^2\eta^2L^2\sigma^2)$$

Combine everything together, we have for equation 11:

$$\left\langle \nabla f(\bar{x}_t), \frac{1}{K}\sum_{k=1}^{K} \nabla f(x_t^{(k)}; \mathcal{B}_t^{(k)}) \right\rangle$$

$$\geq \frac{1}{4}\mathbb{E}\Big[\big\|\nabla f(\bar{x}_{t-\tau})\big\|^2\Big] - 2\nu^2\sigma^2 - 128\tau\eta^2 L^4 \sum_{\ell=max(t-\tau,0)}^{t-1} \frac{1}{K}\sum_{k=1}^{K}\Big\|x_\ell^{(k)} - \bar{x}_\ell\Big\|^2 - \frac{L^2}{K}\sum_{k=1}^{K}\Big\|x_{t-\tau}^{(k)} - \bar{x}_{t-\tau}\Big\|^2$$

$$- 128\tau\eta^2 L^2 \sum_{\ell=max(t-\tau,0)}^{t-1}\Big\|\nabla f(\bar{x}_\ell)\Big\|^2 - \frac{1}{32}\Big\|\nabla f(\bar{x}_t)\Big\|^2 - 64\tau^2\eta^2 L^2\sigma^2\Big] \tag{14}$$

Finally for the last term in equation 10, we have:

$$\frac{\eta^2 L}{2}\|\bar{d}_t\|^2 = \frac{\eta^2 L}{2}\|\frac{1}{K}\sum_{k=1}^{K}\nabla f(x_t^{(k)}; \mathcal{B}_t^{(k)})\|^2 \leq \eta^2 L\|\frac{1}{K}\sum_{k=1}^{K}\nabla f(x_t^{(k)})\|^2 + \eta^2 L\sigma^2$$

$$\leq 2\eta^2 L^3 \frac{1}{K}\sum_{k=1}^{K}\|x_t^{(k)} - \bar{x}_t\|^2 + 2\eta^2 L\|\nabla f(\bar{x}_t)\|^2 + \eta^2 L\sigma^2 \tag{15}$$

Plug equation 14 and equation 15 to equation 10, we have:

$$\mathbb{E}[f(\bar{x}_{t+1})] \leq \mathbb{E}\Big[f(\bar{x}_t) - \frac{\eta}{4}\mathbb{E}\Big[\big\|\nabla f(\bar{x}_{t-\tau})\big\|^2\Big] + 2\eta\nu^2\sigma^2 + 64\tau^2\eta^3 L^2\sigma^2 + \eta^2 L\sigma^2$$

$$+ 128\tau\eta^3 L^2 \sum_{\ell=max(t-\tau,0)}^{t-1}\Big\|\nabla f(\bar{x}_\ell)\Big\|^2 + (4\eta^2 L + \frac{\eta}{32})\Big\|\nabla f(\bar{x}_t)\Big\|^2$$

$$+ 4\eta^2 L^3 \frac{1}{K}\sum_{k=1}^{K}\|x_t^{(k)} - \bar{x}_t\|^2 + \frac{\eta L^2}{K}\sum_{k=1}^{K}\Big\|x_{t-\tau}^{(k)} - \bar{x}_{t-\tau}\Big\|^2$$

$$+ 128\tau\eta^3 L^4 \sum_{\ell=max(t-\tau,0)}^{t-1} \frac{1}{K}\sum_{k=1}^{K}\Big\|x_\ell^{(k)} - \bar{x}_\ell\Big\|^2\Big] \tag{16}$$

In this last step we bound $\|\nabla f(\bar{x}_s)\|^2, t - \tau \leq s \leq t$ *w.r.t.* $\|\nabla f(\bar{x}_{t-\tau})\|^2$. For any $s \in [t - \tau, t]$:

$$\|\nabla f(\bar{x}_s)\|^2 \leq 2\|\nabla f(\bar{x}_{t-\tau})\|^2 + 2\|\nabla f(\bar{x}_s) - \nabla f(\bar{x}_{t-\tau})\|^2$$

$$\leq 2\|\nabla f(\bar{x}_{t-\tau})\|^2 + 2\tau L^2\eta^2 \sum_{\ell=max(t-\tau,0)}^{s-1}\|\bar{d}_\ell\|^2$$

$$\leq 2\|\nabla f(\bar{x}_{t-\tau})\|^2 + 4\tau L^2\eta^2 \sum_{\ell=max(t-\tau,0)}^{s-1}(\frac{1}{K}\sum_{k=1}^{K}\|\nabla f(x_\ell^{(k)})\|^2 + \sigma^2)$$

$$\leq 2\|\nabla f(\bar{x}_{t-\tau})\|^2 + 4\tau L^2\eta^2 \sum_{\ell=max(t-\tau,0)}^{s-1}(\frac{2}{K}\sum_{k=1}^{K}\|\nabla f(x_\ell^{(k)}) - \nabla f(\bar{x}_\ell)\|^2 + 2\|\nabla f(\bar{x}_\ell)\|^2 + \sigma^2)$$

$$\leq 2\|\nabla f(\bar{x}_{t-\tau})\|^2 + 8\tau L^4\eta^2 \sum_{\ell=max(t-\tau,0)}^{t-1} \frac{1}{K}\sum_{k=1}^{K}\|x_\ell^{(k)} - \bar{x}_\ell\|^2$$

$$+ 8\tau^2 L^2\eta^2 \max_{\ell\in[t-\tau,t]}\|\nabla f(\bar{x}_\ell)\|^2 + 4\tau^2 L^2\eta^2\sigma^2$$

By setting $\eta < \frac{1}{4L\tau}$, we have:

$$\max_{\ell\in[t-\tau,t]}\|\nabla f(\bar{x}_\ell)\|^2 \leq 4\|\nabla f(\bar{x}_{t-\tau})\|^2 + 16\tau L^4\eta^2 \sum_{\ell=max(t-\tau,0)}^{t-1} \frac{1}{K}\sum_{k=1}^{K}\|x_\ell^{(k)} - \bar{x}_\ell\|^2 + 8\tau^2 L^2\eta^2\sigma^2$$

Plug this inequality back to equation 16 and used the condition that $\eta < \frac{1}{256L\tau}, \tau \geq 1$:

$$\mathbb{E}[f(\bar{x}_{t+1})] \le \mathbb{E}\Big[f(\bar{x}_t) - \frac{\eta}{16}\mathbb{E}\Big[\big\|\nabla f(\bar{x}_{t-\tau})\big\|^2\Big] + 2\eta\nu^2\sigma^2 + 128\tau^2\eta^3 L^2\sigma^2 + \eta^2 L\sigma^2$$

$$+ 4\eta^2 L^3 \frac{1}{K}\sum_{k=1}^{K}\|x_t^{(k)} - \bar{x}_t\|^2 + \frac{\eta L^2}{K}\sum_{k=1}^{K}\Big\|x_{t-\tau}^{(k)} - \bar{x}_{t-\tau}\Big\|^2$$

$$+ 128\tau\eta^3 L^4 \sum_{\ell=max(t-\tau,0)}^{t-1} \frac{1}{K}\sum_{k=1}^{K}\Big\|x_\ell^{(k)} - \bar{x}_\ell\Big\|^2\Big] \tag{17}$$

This completes the proof of the lemma. $\qquad\square$

**Theorem A.3.** *Under Assumptions 3.3-3.5 and Assumption 4.1, we choose $\eta = \min\left(\frac{1}{256L\tau}, \frac{1}{4LI}, \frac{1}{4c_\nu}, \left(\frac{C_0 C_2}{T}\right)^{1/2}\right)$, for some constants $\tau = O(1/p)$, then with any choice of mini-batch sizes $B \ge 1$, and number of local updates, $I \ge 1$, the iterates generated from Algorithm 2 satisfy:*

$$\frac{1}{T}\sum_{t=1}^{T}\mathbb{E}\|\nabla f(\bar{x}_t)\|^2 \le \frac{C_0}{C_\eta T} + C_1\left(\frac{C_0 C_2}{T}\right) + C_2\left(\frac{C_0 C_2}{T}\right)^{1/2}$$

*where $C_0 = 16(\Delta + \sigma^2/L)$, $C_1 = (32 + 16*128\tau^2 L^2 + 256L^2(I-1)^2\tau)\sigma^2$, $C_2 = 16L\sigma^2$, $C_\eta = \min\left(\frac{1}{256L\tau}, \frac{1}{4LI}, \frac{1}{4}\right)$. $\Delta$ denotes the initial sub-optimality.*

*Proof.* Summing the result of Lemma A.2 for $t = [\tau+1, \tau+T]$ and multiplying both sides by $16/\eta T$ we get:

$$\frac{1}{T}\sum_{t=1}^{T}\mathbb{E}\|\nabla f(\bar{x}_t)\|^2 \le \frac{16\mathbb{E}[f(\bar{x}_{\tau+1}) - f^*]}{\eta T} + 16(2\nu^2 + 128\tau^2\eta^2 L^2 + \eta L)\sigma^2$$

$$+ 32\tau L^2 \frac{1}{T}\sum_{t=1}^{\tau+T}\frac{1}{K}\sum_{k=1}^{K}\Big\|x_\ell^{(k)} - \bar{x}_\ell\Big\|^2$$

$$\le \frac{16\mathbb{E}[f(\bar{x}_{\tau+1}) - f^*]}{\eta T} + 16(2\nu^2 + 128\tau^2\eta^2 L^2 + \eta L)\sigma^2$$

$$+ \frac{128L^2(I-1)^2\tau(T+\tau)}{T}\eta^2\sigma^2$$

where in the second inequality, we use Lemma A.2.

As for $\bar{x}_{\tau+1}$, follow Eq. 10, for $\eta < \frac{1}{L}$ we have:

$$\mathbb{E}[f(\bar{x}_{t+1}) - f(\bar{x}_t)] \le \mathbb{E}\Big[\frac{\eta}{2}\Big\|\nabla f(\bar{x}_t) - \frac{1}{K}\sum_{k=1}^{K}\nabla f(x_t^{(k)}; \mathcal{B}_t^{(k)})\Big\|^2\Big]$$

$$\le \frac{\eta}{2}\frac{1}{K}\sum_{k=1}^{K}\mathbb{E}\Big[\Big\|\nabla f(\bar{x}_t) - \nabla f(x_t^{(k)}; \mathcal{B}_t^{(k)})\Big\|^2\Big]$$

$$\le \eta L^2 \frac{1}{K}\sum_{k=1}^{K}\mathbb{E}\Big[\Big\|\bar{x}_t - x_t^{(k)}\Big\|^2\Big] + \eta\sigma^2$$

Then we sum the above inequality for $t \in [\tau]$ and combine with Lemma A.1 to have:

$$\mathbb{E}[f(\bar{x}_{\tau+1})] - f(x_1) \le 4\eta^3\tau(I-1)^2 L^2\sigma^2 + \eta\sigma^2\tau \le \sigma^2/L$$

where the last inequality follows the condition of $\eta < \frac{1}{256L\tau}$ and $\eta < \frac{1}{4LI}$. Then we have:

$$\frac{1}{T}\sum_{t=1}^{T}\mathbb{E}\|\nabla f(\bar{x}_t)\|^2 \le \frac{16(\Delta + \sigma^2/L)}{\eta T} + 16(2\nu^2 + 128\tau^2\eta^2 L^2 + \eta L)\sigma^2 + 256L^2(I-1)^2\tau\eta^2\sigma^2$$

where we simplify the last term by $\tau \leq T$. Next, we define $\nu = c_\nu \eta$ for some constant $c_\nu > 0$, and the constant $C_0 = 16(\Delta + \sigma^2/L)$, $C_1 = (32c_\nu^2 + 16 * 128\tau^2 L^2 + 256L^2(I - 1)^2\tau)\sigma^2$ and $C_2 = 16L\sigma^2$, then the above inequality can be simplified as:

$$\frac{1}{T} \sum_{t=1}^{T} \mathbb{E}\|\nabla f(\bar{x}_t)\|^2 \leq \frac{C_0}{\eta T} + C_1\eta^2 + C_2\eta$$

By the condition of Lemma A.2, where $\nu < \frac{1}{4}$ and $\eta < \frac{1}{256L\tau}$, and the condition of Lemma A.1, where $\eta \leq \frac{1}{4LI}$, we set $\eta$ to be:

$$\eta = \min\left(\frac{1}{256L\tau}, \frac{1}{4LI}, \frac{1}{4c_\nu}, \left(\frac{C_0 C_2}{T}\right)^{1/2}\right)$$

Suppose we denote $C_\eta = \min\left(\frac{1}{256L\tau}, \frac{1}{4LI}, \frac{1}{4c_\nu}\right)$, then we have:

$$\frac{1}{T} \sum_{t=1}^{T} \mathbb{E}\|\nabla f(\bar{x}_t)\|^2 \leq \frac{C_0}{C_\eta T} + C_1\left(\frac{C_0 C_2}{T}\right) + C_2\left(\frac{C_0 C_2}{T}\right)^{1/2}$$

so the last term is the dominant term for sufficiently large $T$, more precisely, if $T$ satisfies:

$$T \geq \max\left(\frac{C_1^2 C_0}{C_2}, \frac{C_0}{C_\eta^2 C_2^3}\right)$$

the last term is the dominant term. More specially, since $C_\eta = O(\tau^{-1})$ and $C_1 = O(\tau^2)$, we need $T = O(\tau^4)$. Meanwhile, we have $\tau = \tau_{mix}(\nu\pi_{min}) \leq \frac{\ln(N^2\nu^{-1})}{p} = \frac{\ln(N^2\eta^{-1})}{p}$, where we set $c_\nu = 1$ in the last equality, so we have $\tau = O(p^{-1})$. This completes the proof of the theorem. $\square$

## A.2 Proof of Convergence Theorem under Single Node Setting

**Lemma A.4** (Shifted Descent Lemma - Single Node). *For all $t \geq \tau \geq \tau_{mix}(\nu\pi_{min})$, $\nu < \frac{1}{4}$ and $\eta < \frac{1}{256L\tau}$, the one step progress can be bounded as follows:*

$$\mathbb{E}[f(x_{t+1})] \leq \mathbb{E}\left[f(x_t) - \frac{\eta}{16}\mathbb{E}\left[\left\|\nabla f(x_{t-\tau})\right\|^2\right] + \eta\nu^2\sigma^2 + 128\tau^2\eta^3 L^2\sigma^2 + \eta^2 L\sigma^2\right]$$

*Proof.* By the smoothness of $f$, we have:

$$\mathbb{E}[f(x_{t+1})] \leq \mathbb{E}\left[f(x_t) - \eta\left\langle\nabla f(x_t), \nabla f(x_t; \mathcal{B}_t)\right\rangle + \frac{\eta^2 L}{2}\|\nabla f(x_t; \mathcal{B}_t)\|^2\right] \quad (18)$$

For the second term in the above inequality, we split it into three terms:

$$
\begin{aligned}
\left\langle\nabla f(x_t), \nabla f(x_t; \mathcal{B}_t)\right\rangle &= \left\langle\nabla f(x_{t-\tau}), \nabla f(x_{t-\tau}; \mathcal{B}_t)\right\rangle \\
&\quad + \left\langle\nabla f(x_t) - \nabla f(x_{t-\tau}), \nabla f(x_{t-\tau}; \mathcal{B}_t)\right\rangle \\
&\quad + \left\langle\nabla f(x_t), \left(\nabla f(x_t; \mathcal{B}_t) - \nabla f(x_{t-\tau}; \mathcal{B}_t)\right)\right\rangle \quad (19)
\end{aligned}
$$

For the first term in equation 19, we have:

$$
\begin{aligned}
\mathbb{E}\left[\left\langle\nabla f(x_{t-\tau}), \nabla f(x_{t-\tau}; \mathcal{B}_t)\right\rangle\right] &= \mathbb{E}_{\mathcal{F}_{t-\tau}}\left[\left\langle\nabla f(x_{t-\tau}), \mathbb{E}[\nabla f(x_{t-\tau}; \mathcal{B}_t)]\right\rangle\right] \\
&= \frac{1}{2}\mathbb{E}_{\mathcal{F}_{t-\tau}}\left[\left\|\nabla f(x_{t-\tau})\right\|^2 + \left\|\mathbb{E}[\nabla f(x_{t-\tau}; \mathcal{B}_t)]\right\|^2 - \left\|\nabla f(x_{t-\tau}) - \mathbb{E}[\nabla f(x_{t-\tau}; \mathcal{B}_t)]\right\|^2\right] \quad (20)
\end{aligned}
$$

Then we use the property that the distribution of $\mathcal{B}_t$ is close to the stationary distribution conditioned on $\mathcal{F}_{t-\tau}$, i.e. $|\mathbb{P}(z_{t,i} = j) - \pi_j| \leq \nu\pi_j$ when $t \geq \tau \geq \tau_{mix}(\nu\pi_{min})$, then for the difference term in

the above inequality, we have:

$$\mathbb{E}_{\mathcal{F}_{t-\tau}}\left[\left\|\nabla f(x_{t-\tau}) - \mathbb{E}[\nabla f(x_{t-\tau}; \mathcal{B}_t)]\right\|^2\right]$$

$$\leq \mathbb{E}_{\mathcal{F}_{t-\tau}}\left[\frac{1}{B}\sum_{z_{t,i}, i\in[B]}\left\|\sum_{j\in[N]}(\mathbb{P}(z_{t,i}=j) - \pi_j)\nabla f(x_{t-\tau}, j)\right\|^2\right]$$

$$\leq \mathbb{E}_{\mathcal{F}_{t-\tau}}\left[\frac{1}{B}\sum_{z_{t,i}, i\in[B]}\nu^2\sum_{j\in[N]}\pi_j\left\|\nabla f(x_{t-\tau}, j)\right\|^2\right] \overset{(a)}{\leq} \mathbb{E}_{\mathcal{F}_{t-\tau}}\left[2\nu^2\left\|\nabla f(x_{t-\tau})\right\|^2 + 2\nu^2\sigma^2\right]$$

where in (a), we use the bounded bias assumption, plug this back to equation 20 and use the condition that $\nu < \frac{1}{4}$, to have:

$$\mathbb{E}\left[\left\langle\nabla f(x_{t-\tau}), \nabla f(x_{t-\tau}; \mathcal{B}_t)\right\rangle\right]$$

$$\geq \frac{1}{2}\mathbb{E}_{\mathcal{F}_{t-\tau}}\left[\frac{1}{2}\|\nabla f(x_{t-\tau})\|^2 + \left\|\mathbb{E}[\nabla f(x_{t-\tau}; \mathcal{B}_t)]\right\|^2 - 2\nu^2\sigma^2\right]$$

$$\geq \frac{1}{4}\mathbb{E}\left[\|\nabla f(x_{t-\tau})\|^2\right] + \frac{1}{2}\left\|\mathbb{E}[\nabla f(x_{t-\tau}; \mathcal{B}_t)]\right\|^2 - \nu^2\sigma^2 \qquad (21)$$

Next for the second term of equation 19, we have:

$$\left\langle\nabla f(x_t) - \nabla f(x_{t-\tau}), \nabla f(x_{t-\tau}; \mathcal{B}_t)\right\rangle \geq -\left\|\nabla f(x_t) - \nabla f(x_{t-\tau})\right\| \times \left\|\nabla f(x_{t-\tau}; \mathcal{B}_t)\right\|$$

$$\geq -\frac{1}{2}(2\tau\eta^2 L^2\sum_{\ell=max(t-\tau,0)}^{t-1}\left\|\nabla f(x_\ell)\right\|^2 + 2\tau^2\eta^2 L^2\sigma^2 + \left\|\nabla f(x_{t-\tau}; \mathcal{B}_t)\right\|^2)$$

and for the third term in equation 19:

$$\left\langle\nabla f(x_t), (\nabla f(x_t; \mathcal{B}_t) - \nabla f(x_{t-\tau}; \mathcal{B}_t))\right\rangle$$

$$\geq -\left\|\sum_{\ell=max(t-\tau,0)}^{t-1}\eta L\nabla f(x_\ell; \mathcal{B}_\ell)\right\| \times \left\|\nabla f(x_t)\right\|$$

$$\geq -(16\tau\eta^2 L^2\sum_{\ell=max(t-\tau,0)}^{t-1}\left\|\nabla f(x_\ell; \mathcal{B}_\ell)\right\|^2 + \frac{1}{32}\left\|\nabla f(x_t)\right\|^2)$$

$$\geq -(32\tau\eta^2 L^2\sum_{\ell=max(t-\tau,0)}^{t-1}\left\|\nabla f(x_\ell)\right\|^2 + \frac{1}{32}\left\|\nabla f(x_t)\right\|^2 + 32\tau^2\eta^2 L^2\sigma^2)$$

Combine everything together, we have for equation 19:

$$\left\langle\nabla f(x_t), \nabla f(x_t; \mathcal{B}_t)\right\rangle$$

$$\geq \frac{1}{4}\mathbb{E}\left[\|\nabla f(x_{t-\tau})\|^2\right] - \nu^2\sigma^2 - 128\tau\eta^2 L^2\sum_{\ell=max(t-\tau,0)}^{t-1}\left\|\nabla f(x_\ell)\right\|^2 - \frac{1}{32}\left\|\nabla f(x_t)\right\|^2 - 64\tau^2\eta^2 L^2\sigma^2]$$

$$(22)$$

Finally for the last term in equation 18, we have:

$$\frac{\eta^2 L}{2}\|\nabla f(x_t; \mathcal{B}_t)\|^2 \leq \eta^2 L\|\nabla f(x_t)\|^2 + \eta^2 L\sigma^2 \qquad (23)$$

Plug equation 22 and equation 23 to equation 18, we have:

$$\mathbb{E}[f(x_{t+1})] \leq \mathbb{E}\left[f(x_t) - \frac{\eta}{4}\mathbb{E}\left[\left\|\nabla f(x_{t-\tau})\right\|^2\right] + \eta\nu^2\sigma^2 + 64\tau^2\eta^3 L^2\sigma^2 + \eta^2 L\sigma^2\right.$$

$$\left. + 128\tau\eta^3 L^2\sum_{\ell=max(t-\tau,0)}^{t-1}\left\|\nabla f(x_\ell)\right\|^2 + (\eta^2 L + \frac{\eta}{32})\left\|\nabla f(x_t)\right\|^2\right] \qquad (24)$$

In this last step we bound $\|\nabla f(x_s)\|^2, t - \tau \le s \le t$ w.r.t. $\|\nabla f(x_{t-\tau})\|^2$. For any $s \in [t-\tau, t]$:

$$\|\nabla f(x_s)\|^2 \le 2\|\nabla f(x_{t-\tau})\|^2 + 2\|\nabla f(x_s) - \nabla f(x_{t-\tau})\|^2$$

$$\le 2\|\nabla f(x_{t-\tau})\|^2 + 2\tau L^2 \eta^2 \sum_{\ell=max(t-\tau,0)}^{s-1} \|\nabla f(x_\ell; \mathcal{B}_l)\|^2$$

$$\le 2\|\nabla f(x_{t-\tau})\|^2 + 4\tau L^2 \eta^2 \sum_{\ell=max(t-\tau,0)}^{s-1} (\|\nabla f(x_\ell)\|^2 + \sigma^2)$$

$$\le 2\|\nabla f(x_{t-\tau})\|^2 + 4\tau^2 L^2 \eta^2 \max_{\ell \in [t-\tau,t]} \|\nabla f(x_\ell)\|^2 + 4\tau^2 L^2 \eta^2 \sigma^2$$

By setting $\eta < \frac{1}{4L\tau}$, we have:

$$\max_{\ell \in [t-\tau,t]} \|\nabla f(x_\ell)\|^2 \le 4\|\nabla f(x_{t-\tau})\|^2 + 8\tau^2 L^2 \eta^2 \sigma^2$$

Plug this inequality back to equation 24 and used the condition that $\eta < \frac{1}{256L\tau}, \tau \ge 1$:

$$\mathbb{E}[f(x_{t+1})] \le \mathbb{E}\Big[f(x_t) - \frac{\eta}{16}\mathbb{E}\big[\big\|\nabla f(x_{t-\tau})\|^2\big] + \eta\nu^2\sigma^2 + 128\tau^2\eta^3 L^2\sigma^2 + \eta^2 L\sigma^2\Big] \tag{25}$$

This completes the proof of the lemma. $\qquad\square$

**Theorem A.5.** *Under Assumptions 3.3-3.5, we choose $\eta = \min\left(\frac{1}{256L\tau}, \frac{1}{4}, \left(\frac{C_0 C_2}{T}\right)^{1/2}\right)$, for some constants $\tau = O(1/p)$, then with any choice of minibatch sizes $B \ge 1$, the iterates generated from Algorithm 1 satisfy:*

$$\frac{1}{T}\sum_{t=1}^{T}\mathbb{E}\|\nabla f(x_t)\|^2 \le \frac{C_0}{C_\eta T} + C_1\left(\frac{C_0 C_2}{T}\right) + C_2\left(\frac{C_0 C_2}{T}\right)^{1/2}$$

*where $C_0 = 16(\Delta + \sigma^2/L)$, $C_1 = (16 + 16*128\tau^2 L^2)\sigma^2$, $C_2 = 16L\sigma^2$, $C_\eta = \min\left(\frac{1}{256L\tau}, \frac{1}{4}\right)$. $\Delta$ denotes the initial sub-optimality.*

*Proof.* Summing the result of Lemma A.4 for $t = [\tau + 1, \tau + T]$ and multiplying both sides by $16/\eta T$ we get

$$\frac{1}{T}\sum_{t=1}^{T}\mathbb{E}\|\nabla f(x_t)\|^2 \le \frac{16\mathbb{E}[f(x_{\tau+1}) - f^*]}{\eta T} + 16(\nu^2 + 128\tau^2\eta^2 L^2 + \eta L)\sigma^2$$

As for $f(x_{\tau+1})$, follow Eq. 18, for $\eta < \frac{1}{L}$ we have:

$$\mathbb{E}[f(x_{t+1}) - f(x_t)] \le \mathbb{E}\Big[\frac{\eta}{2}\big\|\nabla f(x_t) - \nabla f(x_t; \mathcal{B}_t)\big\|^2\Big] \le \eta\sigma^2$$

Then we sum the above inequality for $t \in [\tau]$ to have:

$$\mathbb{E}[f(x_{\tau+1})] - f(x_1) \le \eta\sigma^2\tau \le \sigma^2/L$$

where the last inequality follows the condition of $\eta < \frac{1}{256L\tau}$. Suppose we denote the initial sub-optimality as $\Delta = f(x_1) - f^*$, then we have:

$$\frac{1}{T}\sum_{t=1}^{T}\mathbb{E}\|\nabla f(x_t)\|^2 \le \frac{16(\Delta + \sigma^2/L)}{\eta T} + 16(\nu^2 + 128\tau^2\eta^2 L^2 + \eta L)\sigma^2$$

Next, we define $\nu = c_\nu \eta$ for some constant $c_\nu > 0$, and the constant $C_0 = 16(\Delta + \sigma^2/L)$, $C_1 = (16c_\nu^2 + 16*128\tau^2 L^2)\sigma^2$ and $C_2 = 16L\sigma^2$, then the above inequality can be simplified as:

$$\frac{1}{T}\sum_{t=1}^{T}\mathbb{E}\|\nabla f(x_t)\|^2 \le \frac{C_0}{\eta T} + C_1\eta^2 + C_2\eta$$

By the condition of Lemma A.4, where $\nu < \frac{1}{4}$ and $\eta < \frac{1}{256L\tau}$, we set $\eta$ to be:

$$\eta = \min\left(\frac{1}{256L\tau}, \frac{1}{4c_\nu}, \left(\frac{C_0 C_2}{T}\right)^{1/2}\right)$$

Suppose we denote $C_\eta = \min\left(\frac{1}{256L\tau}, \frac{1}{4c_\nu}\right)$, then we have:

$$\frac{1}{T}\sum_{t=1}^{T}\mathbb{E}\|\nabla f(x_t)\|^2 \leq \frac{C_0}{C_\eta T} + C_1\left(\frac{C_0 C_2}{T}\right) + C_2\left(\frac{C_0 C_2}{T}\right)^{1/2}$$

so the last term is the dominant term for sufficiently large $T$, more precisely, if $T$ satisfies:

$$T \geq \max\left(\frac{C_1^2 C_0}{C_2}, \frac{C_0}{C_\eta^2 C_2^3}\right)$$

the last term is the dominant term. More specially, since $C_\eta = O(\tau^{-1})$ and $C_1 = O(\tau^2)$, we need $T = O(\tau^4)$. Meanwhile, we have $\tau = \tau_{mix}(\nu\pi_{min}) \leq \frac{\ln(N^2\nu^{-1})}{p} = \frac{\ln(N^2\eta^{-1})}{p}$, where we set $c_\nu = 1$ in the last equality, so we have $\tau = O(p^{-1})$. This completes the proof of the theorem. $\square$

# B   MORE DETAILS ABOUT EXPERIMENTS

## B.1   MORE DETAILS OF FIGURE 1

In Figure 1, we measure the FLOPS rate *vs* Arithmetic Intensity under different data loading speed levels. In particular, the term "arithmetic intensity" comes from the Roofline model (Williams et al., 2009) of the computer systems community. Formally, Arithmetic Intensity is the ratio of total floating-point operations (FLOPS) to the total data movement (Bytes) required to support those FLOPS. The main observation from the figure is: *For a given data loading speeding, a certain arithmetic intensity is necessary to reach the maximum FLOPS rate.*

We perform synthetic experiments to get Figure 1. More specifically, we train a ResNet-18 model to fit the CIFAR-10 dataset on an A5000 GPU. Firstly, we vary the arithmetic intensity by changing the number of gradient steps (denoted as $k$) performed for each sample. In other words, we perform $k$ consecutive gradient steps for each sample loaded to the memory. Note that by the definition of arithmetic intensity, it increases as the value of $k$ increases. As for the data loading speed, we repeat the data loading operation $i$ times per sample to achieve different levels of loading speed. Increase the value of $i$ leads to slower data loading

