# OpenReview forum: "Sharper Analysis of Data Echoing and New Communication-Efficient Algorithm for Data Parallelism"
_ICLR.cc/2025/Conference — Submitted to ICLR 2025_

### Official Review · Reviewer_m2q1 · 2024-10-28

**Soundness:** 2
**Presentation:** 2
**Contribution:** 2
**Rating:** 3
**Confidence:** 2

**Summary:**

This paper provides a sharper analysis of previous data echoing work. The paper concludes that data echoing can get linear speedup proportional to sample reuse times. Then it proposes reducing gradient averaging frequency based on data echoing frequency to reduce communication cost. For evaluation, this paper adopts a cosine diminishing schedule for data echo probability and valid its effectiveness.

**Strengths:**

Theoretical proof of data echoing achieving linear speedup proportional to num of reus times. The authors formulate stochastic data echoing as Markov chain gradient descent problem and provide a sharper analysis.

data echo is a good direction for reducing data loading overhead in distributed environments.

The proposed cosine diminishing schedule on data echoing achieves better model test accuracy.

**Weaknesses:**

1 ) the paper lacks of systematic understanding of how distributed training works and what could be bottleneck. For example, one major contribution in this paper is to reducing cross GPU communication frequency for gradient averaging with probability p^{c}_{t} (detailed in Algorithm2 and section 4). It does not consider how it can be grounded in real world training.

1.1) If every GPU's triggering gradient averaging probability is i.i.d, then it is almost impossible to pre-allocate and pre-form the GPU communication group for each gradient averaging collective. If forming communication groups ad hoc, it means every time before starting communication, we need to initialize a new communication group and ping every involved GPU to build connection, which will be much bigger overhead compared with reduced communication frequency gain.

1.2) If all GPUs communicate at same time but with lower frequency, this kind of techniques already exists as gradient accumulation steps. Further more, comparing with data echoing's reducing data communication frequency with probability and may have model training accuracy loss, gradient accumulation step can mimic identical model training loss curve of pure distributed data parallel (DDP) but communicate gradient at a much lower frequency.

2 ) the paper lacks major results. LLM is a good example for distributed model training. As mentioned in Sec.5, the paper also use wiki text and gpt-2 model training for evaluation. However, I could not find any results in the paper. The only results on cifar-10/100 + small CNN like resnet/mobilenet usually do not need distributed training. Therefore cifar10/100+ small CNN results are not very convincing.

3 ) The paper is lack of novelty. Two major contributions in this paper. First it provides a tighter convergence analysis of previous work of data echoing via formulating stochastic data echoing to markov chain gradient descent. This first contribution is theoretical contribution but does not proposing any new idea. The second contribution is reducing cross-GPU gradient averaging frequency based on data echoing frequency. The idea seems novel in data echoing setting, but there is a much widely adopted and existed approach call gradient accumulation step, which does not hurt model training accuracy at all while reducing gradient averaging communication frequency. One minor novelty is adding cosine diminishing schedule on data echoing, but this novelty contribution is limited, since any diminishing schedule may work in data echoing setting.

minor issues:

all the figures from fig3 to fig7 (especially fig7) in both x and y axis, the texts are too small to see even enlarge to 200%.

**Questions:**

How does communication with some probability compared with widely used gradient accumulation step approach? To me, gradient accumulation step approach does not hurt any model training accuracy loss and much easier to be used in real world applications (i.e. reuse the same communication groups all the time with NCCL/RCCL).

How would this paper's approach works in real distributed training environment? (either larger dataset like imagenet, or larger models like gpt-2/3, llama-2/3)

---

> ### Author Response · Authors · 2024-12-03
>
> Thanks for your review.
>
> For your first concern, we use Pytorch's distributed package for simulation, where the communication group is preallocated. As for how to coordinate the clients, we can either fix the random seed across all clients or simply force the clients to communicate every $I$ steps (set $I =1/p$). We do not claim to invent the local gradient accumulation technique, instead we show the data echoing technique can be combined seamlessly with local gradients without damaging the convergence rate.
>
> For your second concern, please check Figure 7 for our results on GPT-2.

---

### Official Review · Reviewer_G1Ro · 2024-10-31

**Soundness:** 2
**Presentation:** 2
**Contribution:** 3
**Rating:** 5
**Confidence:** 3

**Summary:**

The paper suggests a markovian perspective when analysing the data echoing algorithm, the technique that allows to mitigate the data loading overhead. Data echoing combined with Stochastic Gradient Descent uses the biased gradient in the parameter update. The authors show the boundedness of such bias under mild assumptions and run a few experimental runs to show its efficiency in practice. In addition to that, they introduce a novel data echoing algorithm that is adjusted to data parallelism setting.

**Strengths:**

The authors tackle more general setting than the prior works by considering non-convex optimization problems. They take an interesting approach to model data echoing in SGD through Markov chain processes. They provide an analysis of the problem, proving the comparable convergence to SGD while having a linear speedup in terms of number of reuse steps.

The authors also provide an adaptation of the data echoing algorithm to a data parallelism setup. They noticed that the communication that happens when synchronizing the weights negates the benefits that data echoing introduces for a single node setup. Thus, they combine a data echoing algorithm with the delayed weight synchronization to keep idle time low, while proving the convergence of the novel optimization scheme.

Overall, the work extends on the previous line of work with an interesting algorithmic and theoretical contributions.

**Weaknesses:**

As a person who has not been familiar with data echoing before, I find the experimental results counter-intuitive and misleading. In particular, they do not answer the question "if your hardware has a particular data loading speed what level of data echoing you need to use?".

I see the data echoing technique as a way to reduce the idle time that is due to the long waiting for the next loaded samples. So I expect data echoing may decrease the total training time, but as a method that uses a biased gradient estimation, it should require more optimization steps to converge to a minimum compared to a vanilla SGD. In the experimental section we only see experiments with respect to «number of example loads», which creates an impression that data echoing is in general always better than SGD. This paper lacks more figures that are plotted against the number of gradient steps or epochs to see a bigger picture and compare convergences. More thorough experimental section would provide a better intuition whether you need to use data echoing or not if your hardware support a certain data transfer speed.

Moreover, to have a fair comparison of the data echoing algorithm and SGD, we need to choose the best learning rate for each algorithm independently, right now the learning rate is the same for both algorithms. Indeed, if roughly speaking data echoing performs as an SGD that reuses the same batch several times, then the similar effect can be sometimes obtained by increasing the learning rate for SGD, which can be partially observed on some figures.

Concerning the theoretical analysis, the convergence results look reasonable, but the proofs are not easy to follow as some derivation steps are either skipped or not explained, which makes it harder to read and thus verify the correctness of the proofs for a person not familiar with this line of work. Overall, the appendix seems to be written hastly, I suggest the authors to pay more attention to how they explain their derivations and add missing details in the proofs.

Minor remarks:

line 038: parallelism -> parallel

line 237: slow -> slows

line 265: it is stated that $\nabla f(x_t, B_t)$ approximates well $\nabla f(x_{t-\tau})$ and that it follows from Lemma 3.6. Please comment more on this as it is not a straightforward induction

line 307: explain how do you get the minimum burn-in time (a lower bound for T)?

line 363: effect to -> effect on

line 370: $c_{\nu}$ was never introduced before

Use $\times$ or $\cdot$ for multiplication, instead of *

Figure 1 from the main paper contradicts the description in the appendix, as in one place higher i means higher data loading speed, in other place the opposite.

I suggest using separate numbering for Definition, Lemmas and Theorem independent of section number, so that there are Theorem 1 and Theorem 2, instead of Theorem 3.8 and Theorem 4.2

In equation 4, I would use $p_t$ instead of $p$ directly to simplify Algorithm 1 and 2 descriptions (e.g. line 3 of Algorithm 1)

The order of Figures doesn’t follow their order of mentioning in the text

**Questions:**

Can you do an experiment where each algorithm is compared with their best corresponding learning rate? (see above)
How to set your algorithm for a given data loading speed for it to do the best performance?

Please provide the modern GPU performance numbers for a standard SGD algorithm and what it corresponds to in figure 1?

The equation 8 in the appendix is explained with independent sampling property, which seems to contradict the whole Markov Chain formulation of the optimization problem where $d_l$ depends on $d_{l-1}$ due to data echoing. Can you please explain these transitions more in detail?

---

> ### Author Response · Authors · 2024-12-03
>
> Thanks for your review.
>
> For your first concern, we use the "number of example loads" based on the assumption that data loading is the primary bottleneck. In other words, the data loading operation accounts for the majority of the training time, allowing us to use the number of example loads as a proxy for **training time**. If we were to use the number of gradient steps instead, the data echoing would appear slower than standard SGD due to the bias introduced by the sampling process. However, this comparison would not provide meaningful insights into which algorithm runs faster.
>
> For your second concern, in our experiments, we compare data echoing and sgd under the same learning rate, and results show that our algorithm (with a proper schedule for data loading) outperforms sgd under both small and large learning rates.
>
> For your third concern, we will polish our theoretical proof and add more explanations and intermediate steps in our final version. *In particular, equation 8 does not need independent sampling property, instead we use the triangle inequality, we will correct it.*

---

### Official Review · Reviewer_gxue · 2024-11-01

**Soundness:** 2
**Presentation:** 3
**Contribution:** 2
**Rating:** 3
**Confidence:** 2

**Summary:**

This paper provides a convergence analysis for SGD with reused data samples (i.e., dada echoing). The analysis is standard for non-convex optimization; however, it uses an unusual assumption (line 076) that the gradient is still unbiased with the reused data samples.

**Strengths:**

1. The work is well-motivated: As data movement is expensive in modern computer systems, data reusing can significantly improve the performance of neural network training. Previous work has shown practical benefits and preliminary theoretical results for reusing data samples in SGD. This work aims to provide a sharper analysis for the data echoing algorithm.

2. Presentation is good: The paper is well written. The author clearly described the data echoing algorithms and their theoretical results.

**Weaknesses:**

My main concern about the paper is the unusual assumption it uses (line 076). The key difference between standard SGD and data echoing SGD lies in gradient computation. For standard SGD, we can assume unbiased gradients due to i.i.d. sampling of data. However, for data echoing SGD, I strongly suspect this assumption doesn't hold. If we could make the same assumption for data echoing, I don't see how the analysis would differ from standard SGD.

**Questions:**

You assume the gradient is unbiased for large enough $\tau$; however, in the actual algorithm, I guess $\tau$ is limited by M. Can you give more explanation on the assumption?


----
It seems the discussion period has ended -- The authors posted responses at the last minute which does not give time for thorough discussion.

The last response from the authors is interesting. How could you say something is large enough with big O notation, shouldn't it be $\Omega$. At this point, I feel  that either I have a serious misunderstanding of the paper, or there are serious errors in the paper. Too bad we don't have time for sufficient discussion. I will leave it to the AC and other reviewers for the final decision.

---

> ### Author Response · Authors · 2024-12-03
>
> Thanks for your review.
>
> We want to clarify that we **NEVER** assume  i.i.d. sampling in our analysis. Line 076 in the introduction section is just an intuitive explanation of our proof idea: the correlation between to samples diminish as their distance increase in a Markov chain (which is the core of the **mixing time** concept). Our proof is based on viewing the example sequence in data echoing as a Markov chain.
>
> Please let us know for anything unclear.

---

> > ### Comment · Reviewer_gxue · 2024-12-03
> >
> > I want to clarify that I **NEVER** said you assumed i.i.d sampling. My question was: Line076 basically says gradient is unbiased, which is reasonable with i.i.d sampling, but it’s unclear to me why it can be true with data echoing.

---

> > > ### Author Response · Authors · 2024-12-03
> > >
> > > Thanks for clarifying. As we stated, this is just for illustration purpose and is not an assumption, for more rigorous statement, we should change the narratives in line076 as $E[\nabla f(x_{t}^{(m)}; B_{t+\tau})| B_{t}] \approx \nabla f(x_{t}^{(m)})$. The intuition is that as $\tau$ increases, the correlation between $B_t$ and $B_{t+\tau}$ diminishes and the distribution of $B_{t+\tau}$ is almost the stationary distribution. Please check line800-808 in the appendix for the actual use of this idea in the proof.

---

> ### Author Response · Authors · 2024-12-03
>
> The statement starting from line074 can be adjusted as : In contrast, we perform a different analysis by bounding $\|\nabla f(x_{t+\tau}^{(m)}; B_{t+\tau}) - \nabla f(x_{t}^{(m)})\|$. Given that $B_{t+\tau}$ is almost independent from $B_{t}$ for sufficiently large $\tau$, $E[\nabla f(x_{t}^{(m)}; B_{t+\tau})|B_t]\approx \nabla f(x_{t}^{(m)})$. If $x_{t}^{(m)}$ is close to $x_{t+\tau}^{(m)}$, we can bound $\|\nabla f(x_{t+\tau}^{(m)}; B_{t+\tau}) - \nabla f(x_{t}^{(m)})\|$ given the function $f$ is smooth.

---

> ### Comment · Reviewer_gxue · 2024-12-03
>
> Okay, so back to the question in my original review. $tau$ is limited by M. How can you assume gradient variance diminishes as $tau$ goes infinity?

---

> ### Author Response · Authors · 2024-12-03
>
> We **DON'T** assume nor need $\tau$ goes infinity. Our condition to $\tau$ is shown in Theorem 3.8 which is $\tau = O(1/p)$ (this is what we call sufficiently large). Please read the appendix for more details of the proof.

---

> > ### Comment · Area_Chair_DtPN · 2024-12-03
> > **Please keep the discussion civil**
> >
> > Dear Authors and Reviewers,
> >
> > Please keep the discussion civil. OpenReview is accessible to the public, and author identities become non-anonymous after the review period. I encourage everyone to edit their responses to make them more appropriate for public viewing.

---

### Official Review · Reviewer_XCpD · 2024-11-03

**Soundness:** 3
**Presentation:** 3
**Contribution:** 3
**Rating:** 5
**Confidence:** 3

**Summary:**

This paper proposes a new analysis (ie extending and improving previous work) of data echoing, an important technique use in practice in the training of DNN.
Moreover, the author also propose a new (communication efficient) technique tackling the problem of communication bottleneck.
 Numerical experiments support the proposed techniques.

**Strengths:**

- The paper is well written, easy to follow and the contributions are stated clearly.
- the paper proposes an improvement/extension over the work of  Agarwal et al. (2020) by extending their results to the non-convex setting and do not require the gradients to be bounded. Rather, they propose require an assumption present from in Even, 2023. I have not written the proof in details, but I am not surprised by this result.
- to the best of the reviewer's knowledge the problem of finding  "communication efficient data echoing algorithm" has not been addressed in the litterature. The reviewer this is in an interesting direction worth tackling (which this paper does)

**Weaknesses:**

- the reviewer finds the idea of the cosine scheduler for data loading probability particularly interesting.
However, the reviewer wishes to see some theoretical/formal arguments on the soundness of this technique/how this impacts convergence/the results developed in the paper.
Perhaps this is trivial (admittedly the reviewer is not an expert on this topic). In any case, the reviewer does believe this should be stated/clarified.
- The numerics should be more extensive/have more results. MobileNet-V2 are not SOTA anymore (see MobileNet-V3). Moreover, the application is for the training of neural nets. Hence the reviewer expects that results for more modern/SOTA architectures  (transformers....) should be present, regardless of the speed of these architectures at inference time.
- please edit the graphs in figure 4/5 to include the name of the dataset, model and lr (as title for instance....)

I am for now putting a 5/ marginally below acceptance threshold in order to have those few comments addressed.
I would be open to increase my score provided those points are properly addressed (especially about the numerics)

**Questions:**

- please see my comment above on cosine LR schedulers and the comment on the architectures used for the numerical experiments
- the reviewer is curious how definition 3.1 differs from the definition of a "standard" markov chain.
Note that i am not raising this in the "weaknesses" section but I do believe this could be clarified/highlighted.

---

> ### Author Response · Authors · 2024-12-03
>
> Thanks for your review.
>
> **Cosine scheduler**: In our theoretical analysis (Theorem 3.8), we show for any given $p$ (probability of loading new samples), there exists a constant $T_0$, such that for $T > T_0$, data echoing has linear speed up w.r.t. $p$, meanwhile $T_0$ is inverse proportionally to $p$. In other words, for smaller $p$ values, we need to wait longer (larger $T$) to witness the acceleration effect, this inspires us to adopt a diminishing schedule for $p$. In our experimental section, we test different schedules including cosine, linear and multi-step diminishing schedule (Figure 3), and find the cosine schedule has the best performance.
>
> **Numerical results**: We indeed include numerical results beyond neural nets, as shown by the language modeling task in Figure 7, we consider the Wikitext-2 dataset over the GPT-2 model.
>
> **Definition 3.1**: Definition 3.1 is a standard definition to a finite-state time-homogeneous Markov Chain.

---

### Comment · Area_Chair_DtPN · 2024-11-21
**No author response yet**

Dear Submission12780 Authors,

ICLR encourages authors and reviewers to engage in asynchronous discussion up to the 26th Nov deadline. It would be good if you can post your responses to the reviews soon.

---

### Meta-Review · Area_Chair_DtPN · 2024-12-09

**Metareview:**

The paper proposes a theoretical analysis of the "data echoing" setting, in which data is reused during training instead of waiting for a new batch of samples. The paper also proposes a new algorithm for the data echoing setting.

Reviewers agreed that the data echoing setting was relevant and worth investigating. However, multiple reviewers raised concerns about the technical quality of the theoretical analysis, including unclear assumptions and difficult-to-follow proofs. One reviewer also raised concerns that the proposed method does not take into account real-world behavior of distributed GPU systems, and introduces new empirical bottlenecks that may outweigh the theoretical gains.

**Additional Comments On Reviewer Discussion:**

Reviewers raised concerns about the technical quality of the theoretical analysis, including unclear assumptions and difficult-to-follow proofs. The author rebuttal did not convince reviewers that the concerns were fully addressed, and the paper would probably benefit from a careful rewriting pass.

One reviewer also raised concerns that the proposed method does not take into account real-world behavior of distributed GPU systems. This concern was not addressed by the author rebuttal.

---

### Decision · Program_Chairs · 2025-01-22

Reject